



# The importance of turbulent ocean-sea ice nutrient exchanges for simulation of ice algal biomass and production with CICE6.1 and Icepack 1.2

Pedro Duarte[1], Philipp Assmy[1], Karley Campbell[2,3], Arild Sundfjord[1]

[1] Norwegian Polar Institute, Fram Centre, Tromsø, Norway

[2] Department of Arctic and Marine Biology, UiT The Arctic University of Norway, Norway

[3] Bristol Glaciology Centre, University of Bristol, UK

*Correspondence to*: Pedro Duarte (Pedro.Duarte@npolar.no)

**Abstract.** Different sea-ice models apply unique approaches in the computation of nutrient diffusion between the ocean and the ice bottom, which are generally decoupled from the calculation of turbulent momentum and heat flux. Often, a simple molecular diffusion formulation is used. We argue that nutrient transfer from the ocean to sea ice should be as consistent as possible with momentum and heat transfer, since all these fluxes respond to varying forcing in a similar fashion. We hypothesize that biogeochemical models which do not consider such turbulent nutrient exchanges between the ocean and the sea-ice underestimate bottom-ice algal production. The Los Alamos Sea Ice Model (CICE + Icepack) was used to test this hypothesis by comparing simulations with molecular and turbulent diffusion of nutrients into the bottom of sea ice, implemented in a way that is consistent with turbulent momentum and heat exchanges. Simulation results support the hypothesis, showing a significant enhancement of ice algal production and biomass when nutrient limitation was relieved by bottom-ice turbulent exchange. Our results emphasize the potentially critical role of turbulent exchanges to sea ice algal blooms, and the importance of thus properly representing them in biogeochemical models. The relevance of this becomes even more apparent considering ongoing trends in the Arctic Ocean, with a predictable shift from light to nutrient limited growth of ice algae earlier in the spring, as the sea ice becomes more fractured and thinner with a larger fraction of young ice with thin snow cover.

## 1 Introduction

Momentum, heat and mass fluxes between the ocean and the sea-ice are of utmost importance to predict sea-ice motion, thermodynamics, and biogeochemistry. Considering the interlinks between these processes one would expect that sea-ice models used a common approach to compute them, notwithstanding their obvious specificities. However, when we look at models published over the last decades, we find not only inter-model differences in the physical concepts used to describe the processes responsible for some of the above fluxes, but also intra-model differences in the approaches used in calculating, for example, heat and mass fluxes. In this work we will focus on the differences related with the vertical diffusion of tracers





between the water column and the bottom-ice and attempt to explore their consequences on nutrient limitation for sea-ice algal
growth.
The most common processes found in the literature to model nutrient exchanges between the water and the sea ice are based
on entrapment during freezing, release during melting and diffusive or convective fluxes (e.g. Arrigo et al., 1993; Jin et al.,
2006; Tedesco and Vichi, 2010; Jeffery et al., 2011). Arrigo et al. (1993) distinguished nutrient exchanges resulting from
gravity drainage in brine channels, from brine convection in the skeletal layer, dependent on the ice growth rate. These brine
fluxes were used to calculate nutrient exchanges as a diffusive process. Lavoie et al. (2005) also calculated nutrient exchanges
as a diffusive process. Jin et al. (2006; 2008) computed nutrient fluxes across the bottom layer as an advection process
dependent on ice growth rate and based on Wakatsuchi and Ono (1983). Molecular diffusion was also considered. More
recently, other authors have integrated formulations based on hydrostatic instability of brine density profiles, to compute brine
gravity drainage and tracer exchange between the ice and sea water, based on diffusive (Vancoppenolle et al., 2010; Jeffery et
al., 2011) or convective processes (Turner et al., 2013).
Interestingly, the resulting calculation of momentum and heat exchange versus salinity in models is often mismatched. In the
case of the former two, typically, a transfer mechanism (turbulent or not) between the ocean and the sea ice is not dependent
on any type of brine exchange. In the case of salinity, such a mechanism is not considered (e.g. Vancoppenolle et al., 2007;
Turner et al., 2013). Presumably, such differences result from the relative importance of various physical processes for different
tracers. Momentum and heat transfer between the ice and the water are fundamental mechanisms in explaining sea-ice
dynamics and thermodynamics, irrespective of brine exchanges. However, ice desalination depends mostly on brine drainage
and flushing during melt.
Vertical convective mixing of nutrients under the sea ice may result from brine rejection and/or drainage from the sea ice (Lake
and Lewis, 1970; Niedrauer and Martin, 1979; Reeburgh, 1984) and from turbulence due to shear instabilities generated by
drag at the ice-ocean interface (Gosselin et al., 1985; Cota et al., 1987; Carmack, 1986), internal waves and topographical
features (Ingram et al., 1989; Dalman et al., 2019). Gosselin et al. (1985) and Cota et al. (1987) stressed the significance of
tidally induced mixing in supplying nutrients to sympagic algae. Biological demand for silicic acid (hereafter abbreviated as
silicate) and nitrate is limited by the physical supply (Cota and Horne, 1989; Cota and Sullivan, 1990). Vertical nutrient fluxes
between the water and the bottom ice can be calculated from:
$$Fc = -Kz\frac{\Delta C}{\Delta z},\qquad\qquad(1)$$
where $K_Z$ is the vertical eddy diffusivity (m$^2$ d$^{-1}$) and $\Delta C$ is the difference in nutrient concentration over the vertical distance
$\Delta z$ (mmol m$^{-4}$) (Cota et al., 1987).
Even though this eddy diffusion approach was proposed more than 30 years ago, it has been rarely used in the literature. Table
1 summarizes several models published over the last decades and their approaches to the calculation of nutrient diffusion
between the ocean and bottom-ice. Some models do not consider this process and limit nutrient exchanges to brine dynamics
and/or entrapment during freezing and release during melting. Only one of the sampled models (Mortenson et al., 2017) uses





a parameterization based on friction velocity, whereas the others either do not consider nutrient diffusion or use molecular
diffusion.
From this assessment one may divide the ocean-ice exchange processes of existing biogeochemical models into those related
to: (i) entrapment during freezing; (ii) brine drainage, driven by density instability; (iii) flushing, driven by snow and ice
melting, and (iv) diffusive exchanges at the interface between the ocean and the ice, dependent on concentration gradients. In
the absence of ice growth and when brine drainage is limited, diffusive nutrient exchanges between the ocean and the ice have
the capacity to limit primary production. This limitation will be alleviated in the presence of a turbulent exchange mechanism.
We argue that nutrient transfer should be as consistent as possible with momentum and heat transfer since all these fluxes are
closely linked. We hypothesize that models which do not consider turbulent nutrient exchanges between the ocean and the sea-
ice may underestimate bottom-ice ice algal production. Such underestimation will bias the role of sea ice algae in ice associated
food webs and ecosystem services, such carbon dioxide exchanges and their climate feedbacks.
To test the above hypothesis, we use a 1D vertically resolved model and contrast results using the default molecular diffusion
parameterization and a "turbulent" parameterization analogous to that of momentum and heat transfer, based on McPhee

77  (2008).


**Table 1. Model parameterizations used/proposed by different authors to compute diffusion of nutrients at the ice-ocean interface,**
**independent of brine exchanges and/or ice growth/melting. The only example based on friction velocity is that of Mortenson et al.**
**(2017). "None" is used when exchange processes depend solely on brine exchanges and/or ice growth/melting.**

| Source | Type of diffusion | Associated model |
|---|---|---|
| Cota et al. (1987) | Eddy diffusion | - |
| Arrigo et al. (1993) | None | A simulated Antarctic fast ice ecosystem |
| Lavoie et al. (2005) | Molecular diffusion ($1 \times 10^{-9}$ $m^2$ $s^{-1}$) | Ice algal modelling of the Arctic in Resolute Passage, Canadian archipelago. |
| Jin et al. (2006; 2008) | Molecular diffusion according to the authors but using a diffusion coefficient ($1.0 \times 10^{-5}$ $m^2$ $s^{-1}$) that is 4 orders of magnitude higher than molecular diffusion of salt [$1.0 \times 10^{-9}$ $m^2$ $s^{-1}$, following Mann and Lazier (2005)] | Ice-ocean ecosystem model for 1-D and 3-D applications in the Bering and Chukchi seas. |
| Tedesco and Vichi (2010 and e.g. 2019) | None | Biogeochemical flux model in sea ice |
| Vancoppenolle et al. (2010) | None | Modelling brine and nutrient dynamics in Antarctic sea ice |
| Mortenson et al. (2017) | Diffusion parameterized as a function of friction velocity | Biogeochemical model representing the low trophic levels of sea ice and pelagic ecosystems in the Arctic. |
| Hunke et al. (2016) | Molecular diffusion | Los Alamos Sea Ice Model |





## 2 Methods

### 2.1 Concepts

Eq. (1) from Cota et al. (1987) provides the basis for our reasoning about nutrient exchanges between the ocean and the sea-ice bottom being based on a turbulent exchange process, irrespective of other exchanges based on brine dynamics, ice melt and ice growth. Turbulent exchanges may be parameterized through the flux of a quantity at the ocean-ice interface, calculated as the product of a scale velocity and the change in the quantity from the boundary to some reference level (McPhee, 2008):

$$\langle w'S' \rangle = \propto_s u^*(S_w - S_0) \tag{2}$$

Where, $\alpha_s$ is an interface salt/nutrient exchange coefficient (dimensionless); $u^*$ is the friction velocity (m s$^{-1}$); $S_o$ and $S_w$ are sea-ice interface and far-field salinities, respectively.

Hereafter we will assume that salt turbulent exchanges are similar to nutrient exchanges and governed by the same principles and parameters. The main difference between turbulent heat and salt/nutrient exchanges is due to the exchange coefficients that may be higher for heat. The heat exchange coefficient ($\alpha_h$) is around 0.006. The ratio ($R$) between $\alpha_h$ and $\alpha_s$ may vary from unity to a range between 35 and 70 during ice melt and because of double diffusion (McPhee et al., 2008).

The net downward heat flux from the ice to the ocean in the Los Alamos Sea Ice Model (CICE + Icepack) is given by (Hunke et al., 2015) and it is computed according to McPhee et al. (2008) [Eq. (2)]:

$$F_{bot} = -\rho_w c_w \alpha_h u^*(T_w - T_f) \tag{3}$$

Where, $\rho_w$ is the density of seawater (kg m$^{-3}$); $c_w$ is the specific heat of seawater (J kg$^{-1}$ K$^{-1}$); $\alpha_h$ is the heat transfer coefficient (dimensionless); $T_w$ is the water temperature (K); $T_f$ if the freezing temperature (K).

$$[Wm^{-2}] = [kg\ m^{-3}][J\ kg^{-1}\ k^{-1}][dimensionless][m\ s^{-1}][k]$$

We calculate salt or nutrient exchanges using a similar approach:

$$F_N = -\alpha_s u^*(N_w - N_i) \tag{4}$$

In fact, this agrees with McPhee (2008) (see page 112, Fig. 6.3).

$$[g\ m^{-2}\ s^{-1}] = [dimensionless][m\ s^{-1}][g\ m^{-3}]$$

A timescale for this turbulent process may be calculated from:

$$\tau = \frac{\alpha_s u^*}{H}[s^{-1}] \tag{5}$$

Where H is the vertical distance over which diffusion is to be calculated (m). In the Los Alamos Sea Ice Model, it corresponds to the layer thickness of the biogeochemical grid (biogrid) (Jeffery et al., 2017). The above time scale is calculated for consistency with CICE implementation of diffusion, where a comparable time scale is calculated as:

$$\tau = \frac{Dm}{H^2}[s^{-1}] \tag{6}$$

Where $Dm$ is the molecular diffusion coefficient. Eq. (5) or (6) must be multiplied by ice porosity and then used to compute matrix coefficients for the Icepack transport equation along the biogrid (Hunke et al., 2016). Therefore, the implementation of





turbulent diffusion nutrient exchanges in terms consistent with momentum and heat exchanges is quite straightforward depending on changing the timescales from Eq. (6) to (5).

## 2.2 Implementation

We used the Los Alamos Sea Ice Model, which is managed by the CICE Consortium with an active forum (https://bb.cgd.ucar.edu/cesm/forums/cice-consortium.146/) and a git repository (https://github.com/CICE-Consortium). It includes two independent packages: CICE and Icepack. The former computes ice dynamic processes and the latter ice column physics and biogeochemistry. Their development is handled independently with respect to the GitHub repositories (https://github.com/CICE-Consortium). All the changes described below were implemented in two forks to the above repository, one for Icepack and another for CICE and they may be found in Duarte (2021a and b, respectively).

Our simulations may be run using only Icepack, since they are focused on ice column physics and biogeochemistry, without the need to consider ice dynamic processes. However, we used both CICE + Icepack together to allow for use of netCDF based input/output not included in Icepack. Therefore, we defined a 1D vertically resolved model with 1 snow layer and 15 ice layers and 5X5 horizontal cells. This is the minimum number of cells allowable in CICE due to the need to include halo cells (only the central "column" is simulated). Therefore, ice column physics and biogeochemistry were calculated by Icepack but CICE was the model driver. The input file (ice_in) used in this study was included in our CICE fork and it lists all parameters used in the model and described in Hunke et al. (2016), Jeffery et al. (2016), Duarte et al. (2017) and in Tables S1 and S2. Any changes in "default" parameters or any other model settings will be specified.

We made several modifications in CICE to allow using forcing time series collected during the Norwegian Young Sea Ice Expedition (N-ICE2015) (Granskog et al., 2018) and described in Duarte et al. (2017) (see Fig. 2 of the cited authors). These modifications were meant to allow reading of forcing data at higher frequencies than possible with the standard input subroutines in the CICE file ice_forcing.F90.

When the dynamical component of CICE is not used, u* is set to a minimum value instead of being calculated as a function of ice-ocean shear stress (Hunke et al., 2015). Duarte et al. (2017) implemented shear calculations from surface current velocities (one of the models forcing functions) irrespective of using or not the CICE dynamics code. These modifications were also incorporated in the current model configuration so that shear can be used to calculate friction velocity and, thereafter, influence heat and tracer/nutrient exchanges, following Eqs. (3) and (4) and parameters described in McPhee et al. (2008). When the parameter kdyn is set to zero in ice_in, ice dynamics is not computed, but shear is calculated in the modified subroutine icepack_step_therm1, file icepack_therm_vertical.F90. If kdyn is not zero, these calculations are ignored since shear is already calculated in the dynamical part of the CICE code.

A Boolean parameter (Bottom_turb_mix) was added to the input file, which is set to "false" or "true" when the standard molecular diffusion approach or the new turbulent based diffusion approach is to be used, respectively. Another Boolean (Limiting_factors_file) was added to the ice_in file. When set to "true" limiting factor values for light, temperature, nitrogen,





and silicate are written to a text file every model timestep. These are calculated by Icepack biogeochemistry, according to
Jeffery et al. (2016), but there is no writing-output option in the standard code.

## 2.3 Model simulations

Simulations were run for a refrozen lead (RL) without snow cover and for second-year sea ice (SYI) with ~40 cm snow cover
monitored in April-June during the N-ICE2015 expedition (Granskog et al., 2018 and Fig. 1 of Duarte et al. 2017). We ran
simulations with the standard formulations for biogeochemical processes described in Jeffery et al. (2016) and settings
described in Duarte et al. (2017), using mushy thermodynamics, vertically resolved biogeochemistry, and including: brine
drainage, freezing, flushing (Turner et al., 2013, Jeffery et al., 2016) and molecular diffusion as nutrient and algal biomass
exchange mechanisms between the ocean and sea ice. We contrasted the above simulations against others that replaced
molecular diffusion of nutrient exchange at the ice-ocean interface with turbulent diffusion (Table 2), calculated similar to heat
and momentum and following the parameterization described in McPhee et al. (2008) and detailed above. This contrast
provides insight into the effects of changing from molecular to turbulent nutrient diffusion on ice algal production (mg C m$^{-2}$
d$^{-1}$), chlorophyll standing stocks (mg $Chl\ a$ m$^{-2}$) and vertical distribution of chlorophyll concentration (mg $Chl\ a$ m$^{-3}$) [note
that CICE model output for algal biomass in mmol N m$^{-3}$ was converted to mg $Chl\ a$ m$^{-3}$ as in Duarte et al. (2017), using 2.1
mg $Chl\ a$ mmol N$^{-1}$ and following Smith et al. (1993)]. However, due to the concurrent effects of algal biomass exchange
between the ocean and ice, such a contrast is not enough to explicitly test our hypothesis and conclude about the effects of
turbulent -driven nutrient supply on ice algal nutrient limitation. Therefore, simulations were also run contrasting molecular
and turbulent nutrient diffusion, as described above, but restarting from similar algal standing stocks and vertical distributions
within the ice and, switching off algal inputs from the water to the ice. This was done by nullifying the variable algalN, defining
the ocean surface background ice algal concentration, in file icepack_zbgc.F90, subroutine icepack_init_ocean_bio and in the
restart files. In the case of the RL simulations that started with zero ice, first a simulation was run until the 12 May, and then
the obtained ice conditions were used to restart new simulations without algal inputs from the ocean (algalN = 0 mmol N m$^{-3}$
). This way, when the simulations restarted, there was already an ice algal standing stock necessary for the modelling
experiments developed herein. The SYI simulations were, by default, "restart simulations", beginning with observed ice
physical and biogeochemical variables. Therefore, there was already an algal standing stock in the ice from the onset (Text S1
and Table S3).
McPhee et al. (2008) estimated different values for $\alpha_s$ depending on whether the sea ice is growing (highest value) or melting
(lowest value) (Table 2). When running simulations with turbulent bottom diffusion for the RL, in some cases, we used only
the minimum or the maximum values for $\alpha_s$ to allow for a more extreme contrast between molecular and turbulent diffusion
parameterizations. This was done since the former value will tend to minimize differences, whereas the latter will tend to
emphasize them. We also completed simulations for the RL and for SYI changing between the maximum and the minimum
values of $\alpha_s$, when ice was growing or melting, respectively, and following McPhee et al. (2008) (see Table 2 for details). This





parameterization with a variable $\alpha_s$ is likely the most realistic one, accounting for double diffusion during ice melting (McPhee
et al., 2008).
Apart from contrasting the way bottom-ice exchanges of nutrients were calculated, some simulations contrasted different
parameters related to silicate limitation (Table 2). This approach follows Duarte et al. (2017), where simulations were tuned
by changing the Si:N ratio and the half saturation constant for silicate uptake because silicate limitation was leading to an
underestimation of algal growth. From this exercise we were able to assess if such tuning was still necessary after implementing
turbulent diffusion at the ice-ocean interface. Moreover, we repeated simulations with varying snow heights to further
investigate the interplay between light and nutrient limitation under molecular and turbulent nutrient diffusion (Table 2).
Details on model forcing with atmospheric and oceanographic data collected during the N-ICE2015 expedition, including
citations and links to the publicly available datasets are given in Fig. 2 and section 3 of Duarte et al. (2017) and in the
supplementary information. These data sets include: wind speed, air temperature, precipitation, and specific humidity (Hudson
et al., 2015); incident surface short and longwave radiation (Hudson et al., 2016); ice temperature and salinity (Gerland et al.,
2017); sea surface current velocity, temperature, salinity and heat fluxes from a turbulence instrument cluster (TIC) (Peterson
et al., 2016); sea surface nutrient concentrations (Assmy et al., 2016) and sea-ice biogeochemistry (Assmy et al., 2017). Model
forcing files may be found in Duarte (2021c).
















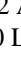



**Table 2. Model simulations. Refrozen lead (RL) simulation RL_Sim1 corresponds to RL_Sim5 described in Duarte et al. (2017) - the simulation leading to a best fit to the observations in that study. The remaining RL simulations 2 – 5 differ from RL_Sim1 in using turbulent diffusion at the ice-ocean interface for nutrients in a comparable way as it is calculated for heat. Moreover, RL_Sim5 differs in the concentration of ice algae in the water column that colonize the sea ice bottom (algalN) and in silicate limitation related parameters. These changes were done iteratively to fit the model to the observations. In RL_Sim2 and RL_Sim3 the maximum ($\alpha_s=0.006$) and the minimum ($\alpha_s=0.006/70=8.6X10^{-5}$) values recommended by McPhee et al. (2008), respectively, are used throughout the simulations, to provide extreme case scenarios for comparison with RL_Sim1. In RL_Sim4, $\alpha_s=8.6X10^{-5}$ when ice is not growing and $0.006$ otherwise, as recommended by McPhee et al. (2008), to account for double diffusive processes during ice melting that slow down mass exchanges. The remaining RL simulations (R__Sim6-9) are like the previous ones (RL_Sim1-4, respectively), except for algalN that was set to zero mmol N m$^{-3}$, and all simulations were restarted with the same values for all variables. Therefore, simulations 6 – 9 may differ only from 13 May 2015, when they were restarted. Second year ice simulation SYI_Sim_1 is based on Duarte et al. (2017) SYI_Sim4 but without algal motion. SYI_Sim2 and SYI_Sim3 use turbulent diffusion at the ice-ocean interface. The former uses a decreased half saturation constant for silicate uptake, just like SYI_Sim1, whereas the latter uses the standard CICE value. The remaining SYI simulations (SYI__Sim4 and 5) are like SYI_Sim1and 2, except for algalN that was set to zero. Simulations SYI_Sim1 and SYI_Sim2 were repeated but with different initial snow thickness of 30, 20 and 15 cm to further investigate the interplay between light and silicate limitation (see text).**

|  | Name | Description |
|---|---|---|
| Refrozen lead simulations | RL_Sim1 | Standard CICE parameters, except for: (i) decreased Si:N ratio from 1.8 to 1.0, well within published ranges (e.g.Brzezinski, 1985; Hegseth, 1992); (ii) decreased half saturation constant for silicate uptake proportionately as the previous ratio, from 4.0 to 2.2 mM within published limits (e.g. Nelson and Treguer, 1992); (iii) decreased colonization in the same proportion as previous parameters by setting ice algal concentration in the water column (algalN) to $11X10^{-4}$ mmol N m$^{-3}$. |
|  | RL_Sim2 | As RL_Sim1 but with molecular diffusivity at the ice-ocean interface replaced with turbulent exchanges, always using the highest value for $a_s$, as recommended by McPhee et al. (2008). |
|  | RL_Sim3 | As RL_Sim2 but always using the lowest value for $a_s$, as recommended by McPhee et al. (2008). |
|  | RL_Sim4 | As RL_Sim3 but using either the lowest value for $a_s$, as recommended by McPhee et al. (2008), when ice is not growing, or the highest one, otherwise. |
|  | RL_Sim5 | As RL_Sim4 but changing half saturation constant for silicate uptake to 5.0 mM and Si:N to 1.7. Moreover, algalN was reduced to $4X10^{-4}$ mmol N m$^{-3}$. |
|  | RL_Sims6-9 | As RL_Sim1-4, respectively, except for algalN that was set to zero, and all simulations were restarted with the same values for all variables in the 13 May 2015. |





| Second year ice simulations | SYI_Sim1 | Standard CICE parameters, except for: (i) the sigma coefficient for snow grain (R_snw) (Urrego-Blanco et al., 2016) that was reduced from 1.5 to 0.8 following Duarte et al. (2017) and (ii) the decreased Si:N ratio and the reduced half saturation constant for silicate uptake and algalN as in RL_Sim1 above |
|---|---|---|
| | SYI_Sim2 | As SYI_Sim1 but with molecular diffusivity at the ice-ocean interface replaced with turbulent exchanges using either the lowest value for $a_s$, as recommended by McPhee et al. (2008), when ice is not growing, or the highest one, otherwise. |
| | SYI_Sim3 | As SYI_Sim2 but changing half saturation constant for silicate uptake back to CICE original value (4.0 mM). |
| | SYI_Sim4 and 5 | As SYI_Sim1and 2, respectively, except for algalN that was set to zero. |

225

## 3. Results

The results of the simulations listed in Table 2 and presented below may be found in Duarte (2021d).

### 3.1 Refrozen lead simulations

All simulations with turbulent diffusion (RL_Sim2 – RL_Sim5, Table 2), predict higher bottom chlorophyll *a* (*Chl a*) concentration than with the standard molecular diffusion formulation (RL_Sim1) (Fig. 1a). Simulations RL_Sim2 - 4 grossly overestimate observations. Simulation RL_Sim3, using the lowest value for $\alpha_s$, is closer both to observations and to RL_Sim1, as well as RL_Sim5, with the latter having the same $\alpha_s$ values of RL_Sim4 but a half saturation constant for silicate limitation increased from its tuned value in Duarte et al. (2017) of 2.2 µM to 5.0 µM and algalN reduced (Table 2) to bring model results closer to observations. Patterns between simulations for the whole ice column and considering both standing stocks and net primary production, are similar to those observed for the bottom-ice (Fig. 1b). Algal biomass is concentrated at the bottom layers (Fig. 2). Concentrations in the layers located between the bottom and the top of the brine network (green lines in the





map plots) are < 10 mg *Chl a* m$^{-3}$. Ice thickness, temperature and salinity profiles are extremely similar among these simulations
(Figs. S1 and S2).
Results for the silicate and nitrogen limiting factors are based on brine concentrations. Limiting factors exhibiting lower values
(more limitation) in RL simulations are silicate, followed by light (Figs. 3, S3 – S5). Limiting values for silicate range between
zero (maximum limitation) and one (no limitation), with higher limitation after May 13 in all simulations (Fig. 3). The most
severe silicate limitation is for RL_Sim1, where values drop to near zero around middle May. Despite the high average bottom
*Chl a* concentration predicted in all simulations the bottom layer is where silicate limitation is less severe after May 13. This
is more evident in simulations with turbulent diffusion, where light limitation at the bottom-ice becomes more severe than
silicate limitation around the end of May (Fig. S6).
Results obtained with RL_Sim6-9, without algal exchanges between the ocean and the ice (see Table 2), show similar patterns
of those observed with RL_Sim1-5, respectively (Fig. 4 versus Fig. 2, Fig. S9 versus Fig. 3, Figs. S7 and S8 versus Figs. S1
and S2, Figs. S10 – S12 versus Figs. S3 – S5).
Interface diffusivity (one of CICE tracers, expressing diffusivity between adjacent biogeochemical layers and between the
bottom layers and the ocean) for simulations with turbulent exchanges are up to two orders of magnitude higher at the bottom
than for simulations with only molecular diffusion (Fig. 5).


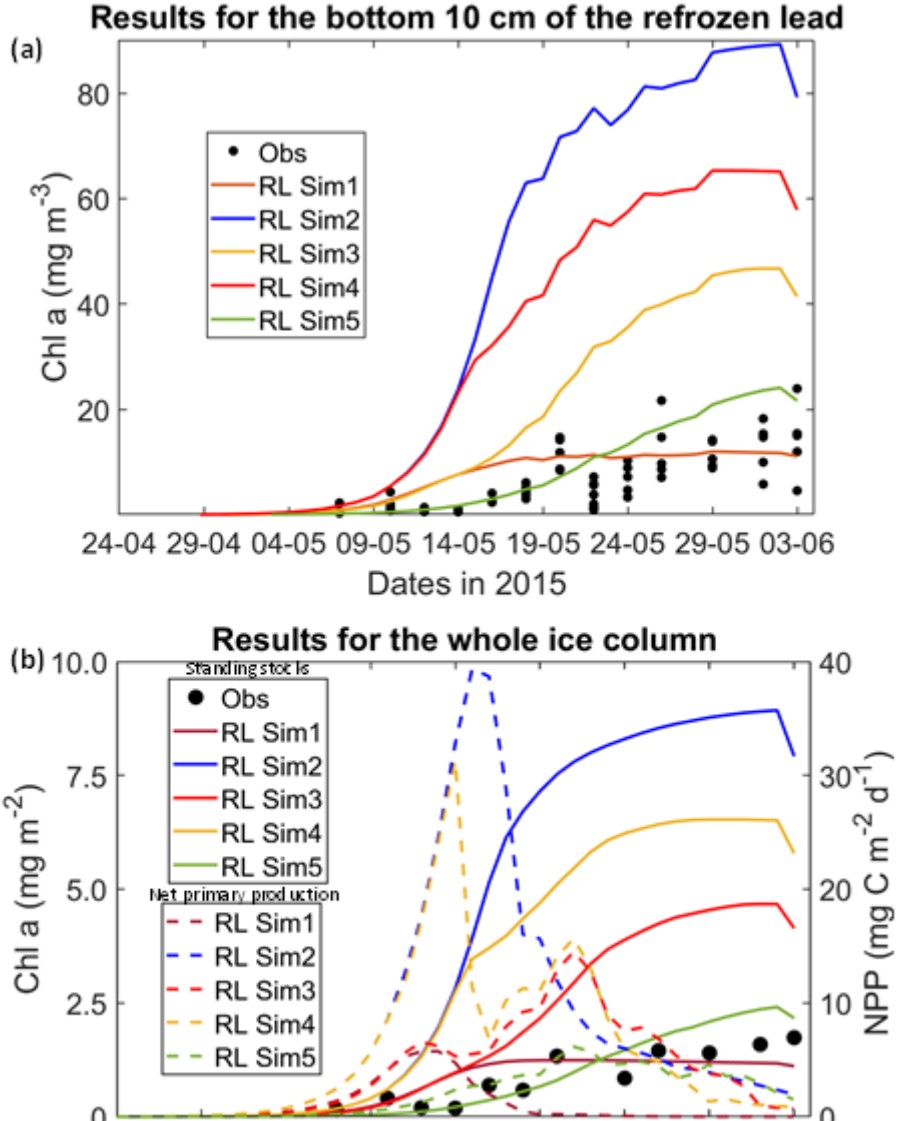

**Figure 1. Daily averaged results for the refrozen lead (RL): (a) Observed and modelled *Chl a* concentration values averaged for the ice bottom 10 cm; (b) Observed and modelled *Chl a* standing stock (continuous lines) and modelled net primary production (NPP) (dashed lines) for the whole ice column (refer to Table 2 for details about model simulations). Observations are the same presented in Duarte et al. (2017).**



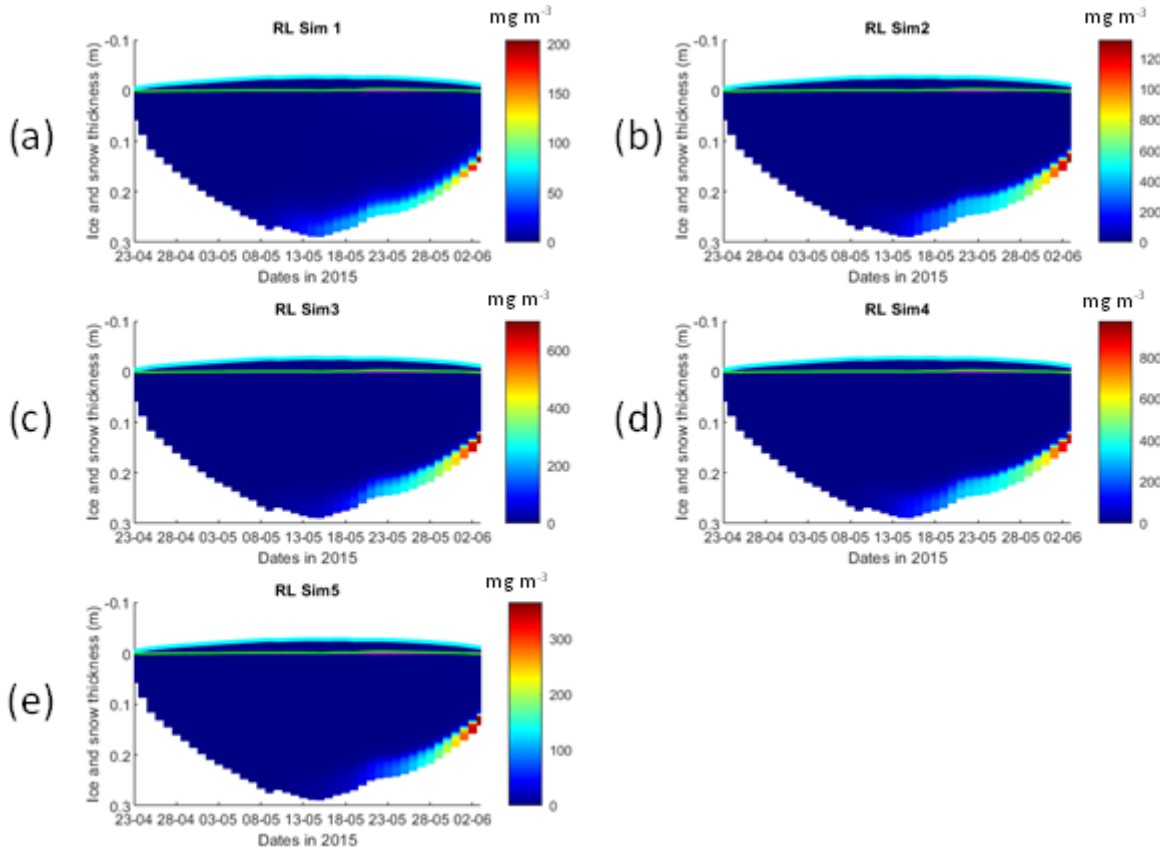

258

**Figure 2. Daily averaged results for the refrozen lead (RL) simulations 1 - 5: Simulated evolution of ice algae *Chl a* as a function of time and depth in the ice (note the colour scale differences between the various panels). Ice thickness is given by the distance between the upper and the lower limits of the maps. The upper regions of the graphs, above the green line with zero values, are above the CICE biogrid and have no brine network. The magenta line represents sea level. Refer to Table 2 for details about model simulations.**



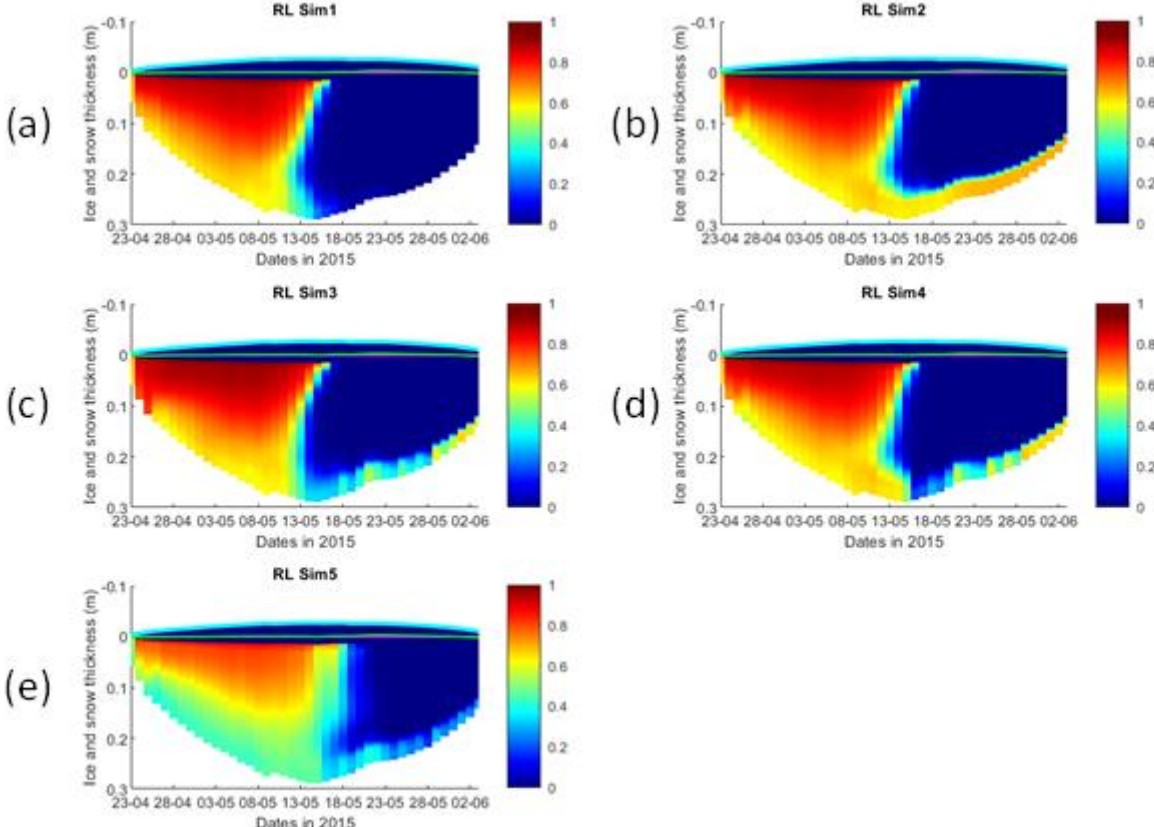

**Figure 3. Daily averaged results for the refrozen lead (RL) simulations 1 - 5: Simulated evolution of silicate limitation (one means no limitation and zero is maximal limitation), as a function of time and depth in the ice. Ice thickness is given by the distance between the upper and the lower limits of the maps. The upper regions of the graphs, above the green line with zero values, are above the CICE biogrid and have no brine network. The magenta line represents sea level. Refer to Table 2 for details about model simulations.**



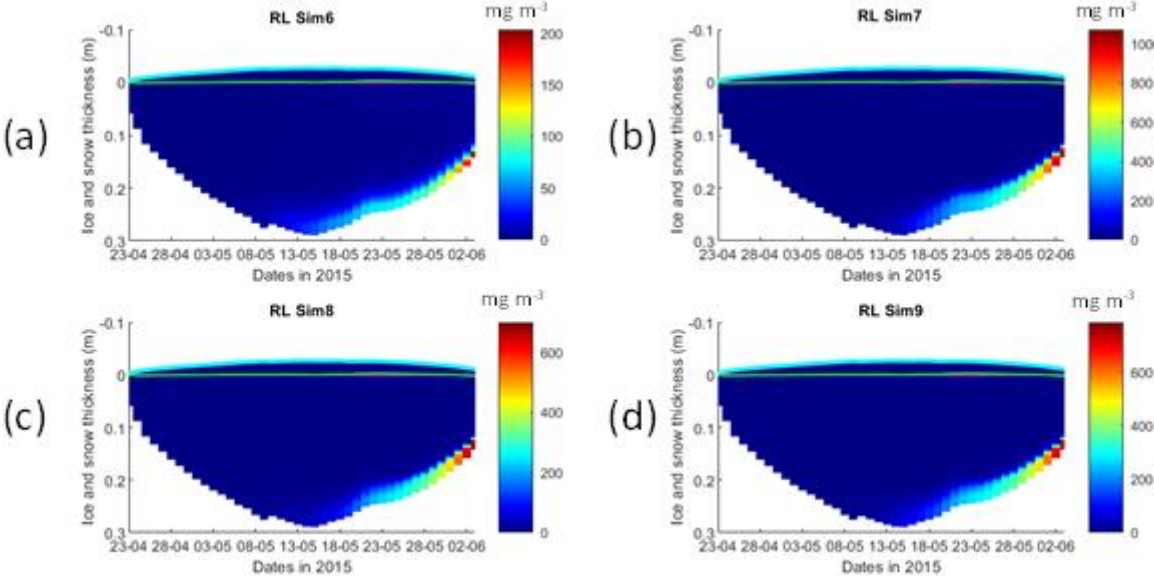

268

**Figure 4. Daily averaged results for the refrozen lead (RL) simulations 6 - 9: Simulated evolution of ice algae *Chl a* as a function of time and depth in the ice (note the colour scale differences between the various panels). Ice thickness is given by the distance between the upper and the lower limits of the maps. The upper regions of the graphs, above the green line with zero values, are above the CICE biogrid and have no brine network. The magenta line represents sea level. Refer to Table 2 for details about model simulations.**



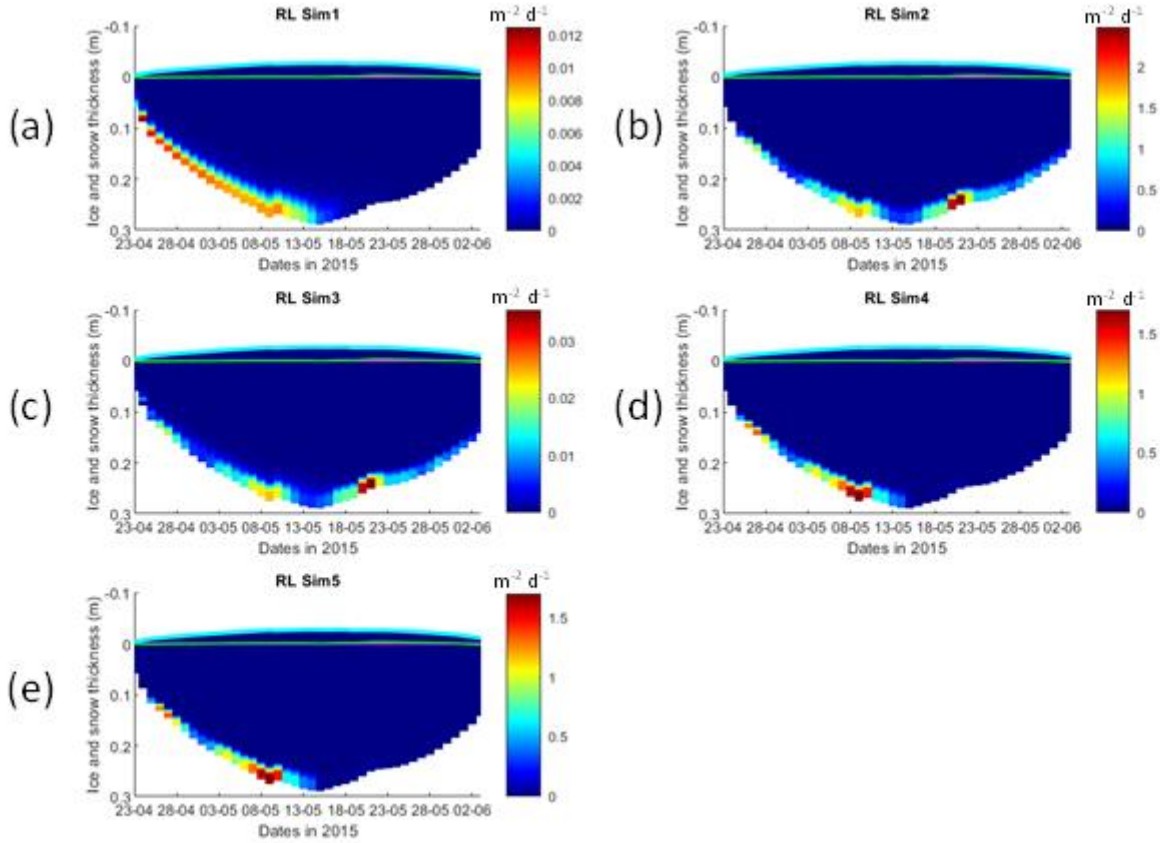

**Figure 5. Daily averaged results for the refrozen lead (RL) simulations 1-5: Simulated evolution of interface diffusivity as a function of time and depth in the ice (note the colour scale differences between the various panels). Ice thickness is given by the distance between the upper and the lower limits of the maps. The upper regions of the graphs, above the green line with zero values, are above the CICE biogrid and have no brine network. The magenta line represents sea level. Refer to Table 2 for details about model simulations.**

## 3.2 Second year ice simulations

Simulations with turbulent diffusion (SYI_Sim2 and 3), predict only slightly higher standing stocks and net primary production than with the standard molecular diffusion formulation (SYI_Sim1) (Fig. 6). The visual fit to the standing stock observations is comparable between the various simulations. Changing the half saturation constant for silicate limitation from 2.2 to 4.0 µM has no impact on model results. This is confirmed by analysing the evolution of *Chl a* concentration as a function of time and depth in the ice (Fig. 7), with only minor differences being apparent towards the end of the simulation, when *Chl a* increases





at the bottom layers in the simulations with turbulent diffusion (SYI_Sim 2 and 3). Ice thickness, temperature and salinity
profiles are extremely similar among these simulations (Fig. S13).
The dominant limiting factor in these simulations is light, seconded by silicate (compare Fig. 8a, c and e with 8b, d and f and
with Fig. S14). Light limitation is less severe after the onset of snow and ice melting at the beginning of June. Silicate limitation
is very strong above the bottom ice. Nitrogen limitation is highest at a depth range between ~0.4 ~0.7 m below the ice top,
with a large overlap with the depth range where a *Chl a* maximum is observed (Fig. 7). Maximum *Chl a* values predicted for
SYI are between two and three orders of magnitude lower than those predicted for the RL (Figs. 2 and 7). However, standing
stocks for the former are larger than those for the latter, considering both observational and model data (Figs. 1b and 6).
Opposite to what was described for the RL simulations, silicate limitation becomes more severe than light limitation at the
bottom layer only in SYI_Sim_1, at the beginning of June, close to the end of the simulation (Fig. S15).
Results obtained without algal exchanges between the ocean and the ice (SYI_Sim4 and 5, see Table 2), show the same patterns
of those observed with SYI_Sim1 and 2, respectively (Fig. 9 versus Fig. 7, Fig. S17 versus Fig. 8, Figs. S18 versus S14a - d
and Figs. S16 versus S13a - d).
Interface diffusivity (one of CICE tracers) for simulations with turbulent exchanges are up to four orders of magnitude higher
at the bottom ice than for simulations with only molecular diffusion (Fig. S19, showing a comparison between SYI_Sim1 and
SYI_Sim2).
SYI_Sim1 and 2 were repeated with varying snow thickness (Table 2 and Figs. 10 and 11). In the former simulation (Fig. 10a),
as snow height decreases, there is a reduction in light limitation and a sharp increase in silicate limitation, overtaking light
limitation (values becoming lower) as early as mid-May. In the latter simulation (Fig. 10b), light limitation prevails irrespective
of snow height, except in the case of the lower snow height of 15 cm where silicate becomes more limiting towards the end of
the simulation. With the decrease in snow height, there is an increase in *Chl a* concentration in all simulations. Highest values
for SYI_Sim2 are ~one order of magnitude larger than those for SYI_Sim1. Moreover, the decrease in snow heights is followed
by an earlier and more intense bottom ice algal bloom.



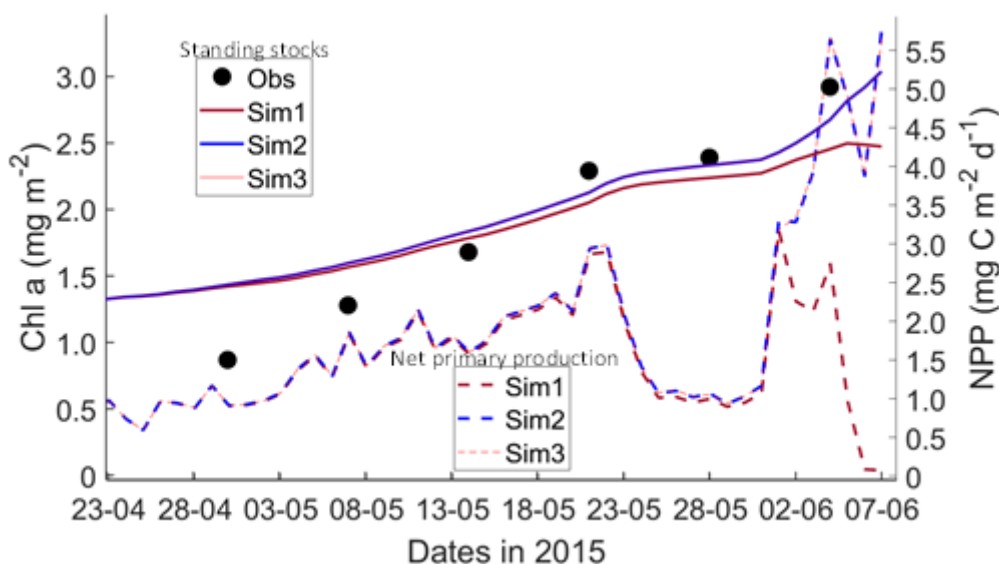

**Figure 6. Daily averaged results for second year ice (SYI) simulations 1 - 3: Observed [same data presented in Duarte et al. (2017)] and modelled *Chl a* standing stock (continuous lines) and modelled net primary production (NPP) (dashed lines) for the whole ice column (refer to Table 2 for details about model simulations).**





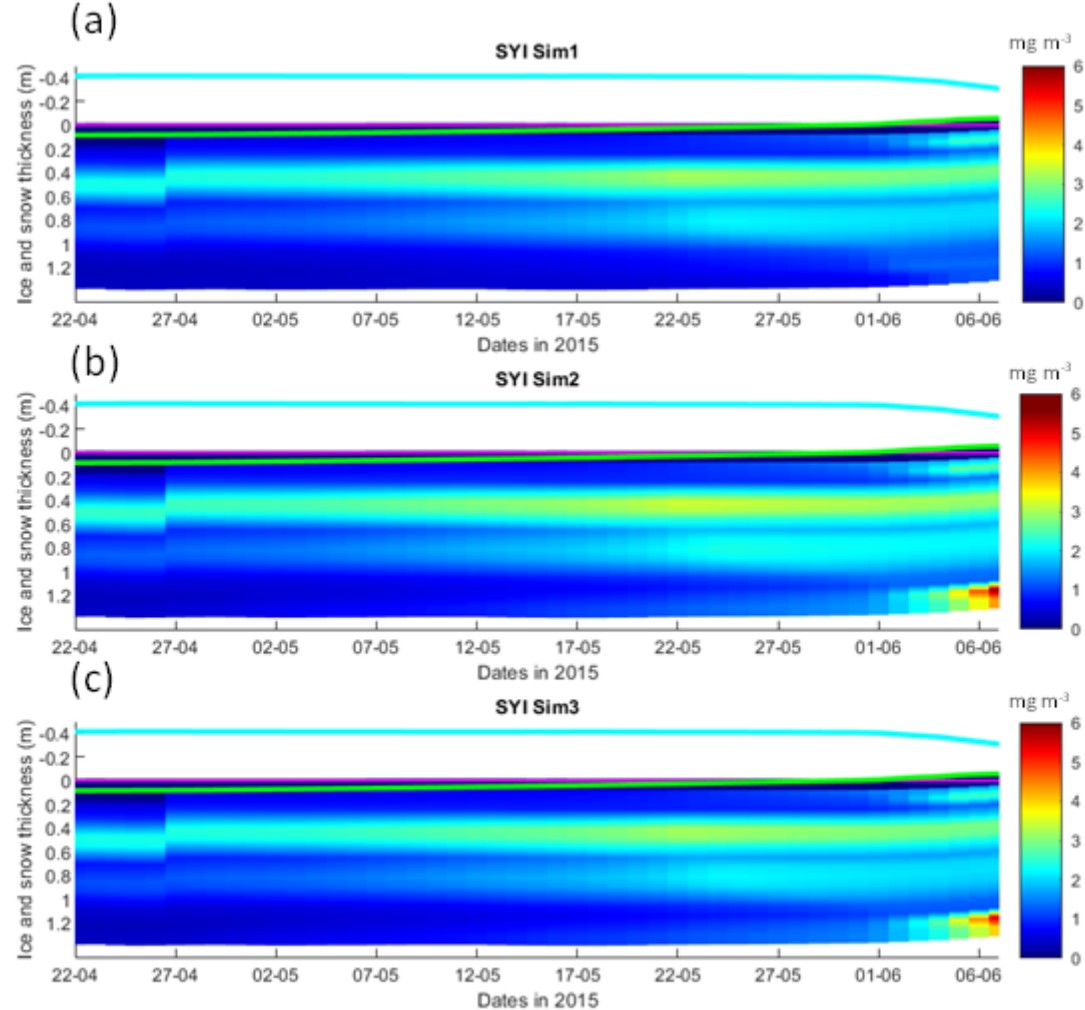

313

**Figure 7. Daily averaged results for second year ice (SYI) simulations 1 - 3: Simulated evolution of ice algae *Chl a* as a function of time and depth in the ice. The upper regions of the graphs, above the green line with zero values, are above the CICE biogrid and have no brine network. The magenta line represents sea level, and the cyan line represents the top of the snow layer. Refer to Table 2 for details about model simulations.**

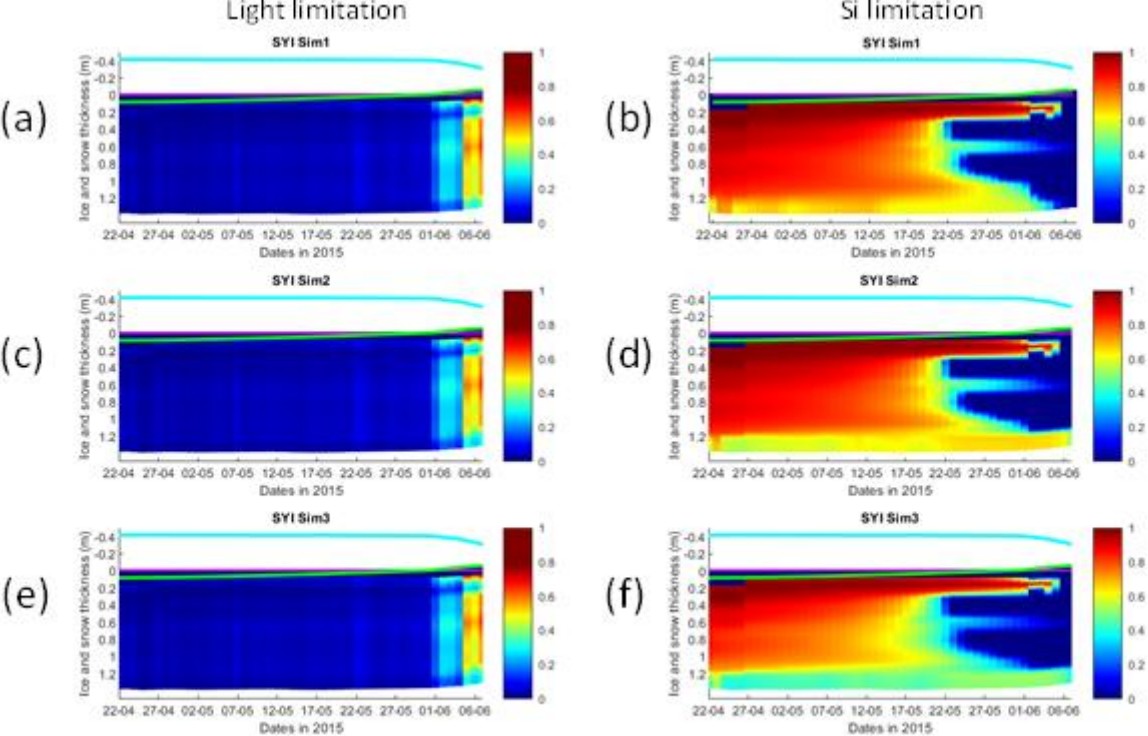

318

**Figure 8. Daily averaged results for second year ice (SYI) simulations 1 - 3: Simulated evolution of light (left panels) and silicate (right panels) limitation (one means no limitation and zero is maximal limitation), as a function of time and depth in the ice. The upper regions of the graphs, above the green line with zero values, are above the CICE biogrid and have no brine network. The magenta line represents sea level, and the cyan line represents the top of the snow layer. Refer to Table 2 for details about model simulations.**

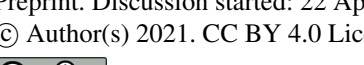

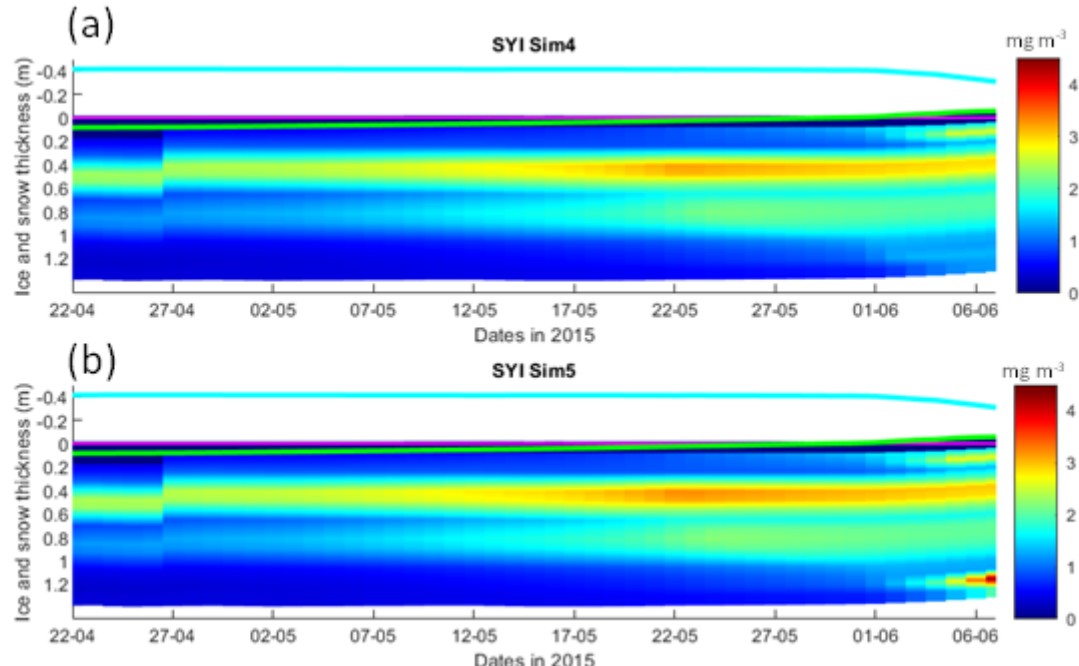

324

**Figure 9. Daily averaged results for second year ice (SYI) simulations 4 and 5: Simulated evolution of ice algae *Chl a* as a function of time and depth in the ice. The upper regions of the graphs, above the green line with zero values, are above the CICE biogrid and have no brine network. The magenta line represents sea level, and the cyan line represents the top of the snow layer. Refer to Table 2 for details about model simulations.**



**Figure 10. Daily averaged results for the second-year ice (SYI) simulations 1 (a) and 2 (b) starting with a snow depth of 40 (default simulation), 30, 20 and 15 cm: Simulated evolution of light (dashed lines) and silicate (continuous lines) limitation (one means no limitation and zero is maximal limitation), as a function of time at the ice bottom layer (one means no limitation). Refer to Table 2 for details about model simulations.**

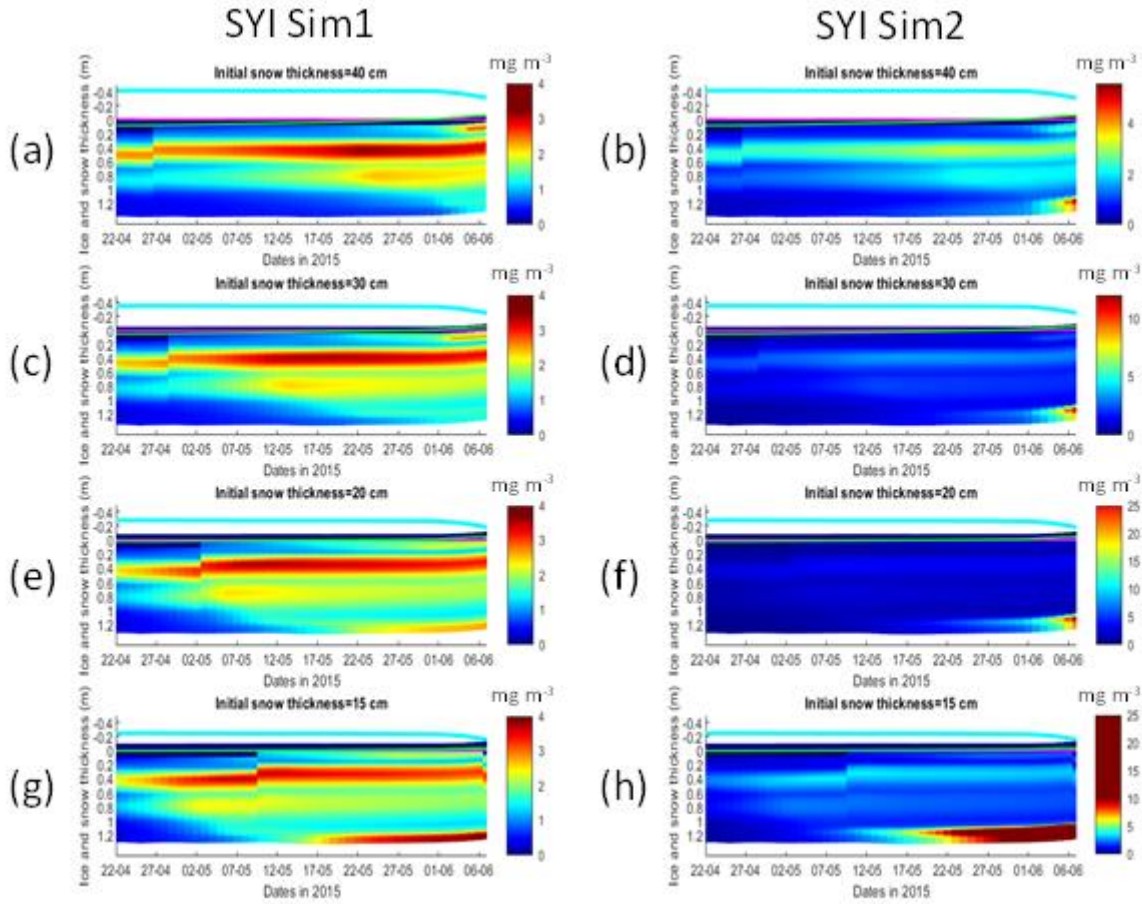

**Figure 11. Daily averaged results for second year ice (SYI) simulations 1 (left panels) and 2 (right panels) starting with a snow depth of 40 (default simulation), 30, 20 and 15 cm: Simulated evolution of ice algae *Chl a* as a function of time and depth in the ice. The upper regions of the graphs, above the green line with zero values, are above the CICE biogrid and have no brine network. The magenta line represents sea level, and the cyan line represents the top of the snow layer. Refer to Table 2 for a description of model simulations.**

## 4.    Discussion

The results obtained in this study support the initial hypothesis, showing that replacing molecular with turbulent diffusion at the ice-ocean interface, formulated in a way consistent with momentum and heat exchanges, leads to a reduction in nutrient limitation that supports a significant increase in ice algal net primary production and *Chl a* biomass accumulation in the bottom ice layers, when production is understood to be nutrient limited.





The implementation of turbulent mixing considerably relieved silicate limitation in the RL simulations, leading to an increase
in bottom *Chl a* concentration and in-ice light absorption, increasing light limitation due to shelf-shading [in the CICE model,
optical ice properties are influenced by ice algal concentrations (Jeffery et al., 2016)].
In the N-ICE2015 biogeochemical dataset (Assmy et al., 2016), the median of dissolved inorganic nitrogen to silicate ratios in
all surface and subsurface water masses, is above 1.7 (unpublished data), which is the upper limit for the nitrogen to silicate
ratio for polar diatoms (e.g. Takeda, 1998; Krause et al. 2018). Therefore, it can be expected that, in the region covered by the
N-ICE2015 expedition, silicate is more limiting than nitrogen for the production yields of the pennate diatoms characteristic
of the bottom-ice communities [the dominant algal functional group in bottom ice, e.g. Leu et al. (2015), van Leeuwe et al.
(2019)]. Elsewhere in the Arctic the opposite may be true, considering nitrate and silicate concentrations presented in Leu et
al. (2015) and the number of process studies documenting such limitation (e.g. Campbell et al. 2016). However, the conclusions
taken here about the effects of turbulent mixing are independent of the limiting nutrient.
Implementing turbulent diffusion has obvious implications for model tuning. Our results for the RL show that with this
formulation it was necessary to increase the half saturation constant for silicate uptake and to reduce the ocean concentration
of algal nitrogen (algalN), reducing the colonization of bottom ice by ice algae, to obtain *Chl a* values comparable to those
observed (RL_Sim5). Therefore, whereas Duarte et al. (2017) had to reduce silicate limitation to improve the fit between
modelled and observational data, the opposite approach was required when using turbulent diffusion. This is an example of
how one can get good model results by the wrong reasons with difficult to predict consequences on model forecasts under
various scenarios.
In the SYI case, only a minor increase in bottom *Chl a* concentration was observed towards the end of simulations SYI_Sim_2
and SYI_Sim_3, when light limitation due to the thick snow cover was relieved by snow melt. Silicate limitation was not as
severe as in SYI_Sim_1, due to greater bottom exchanges in the former simulations. The importance of snow cover in
controlling ice algal phenology has been stressed before [e.g., Campbell et al. (2015), Leu et al. (2015)].
Duarte et al. (2017) used the delta-Eddington parameter, corresponding to the standard deviation of the snow grain size
(R_snow) (Urrego-Blanco et al., 2016), to tune model predicted shortwave radiation at the ice bottom. However, there was
still a positive model bias in June. Therefore, our conclusion about the main limiting role of light in SYI is conservative.
Moreover, in part of SYI cores sampled during the N-ICE2015 expedition, in the period covered by our simulations, with an
unusually high snow thickness (~40 cm), there was no *Chl a* bottom maximum (Duarte et al., 2017; Olsen et al., 2017).
The dominant role of light limitation in SYI was confirmed in the simulations with reduced snow thickness and alleviated light
limitation, with a bottom-ice algal *Chl a* maximum emerging earlier at snow thickness ≤ 20 cm. The reduction of snow heights
had a much larger effect in increasing *Chl a* concentration at the bottom layer when turbulent mixing was used, due to lower
silicate limitation. Reducing snow height led to a relatively early shift from light to silicate limitation when we used molecular
diffusion, whereas this shift occurred only at the very end of the simulated period when we used turbulent diffusion. The effects
of molecular versus turbulent diffusion, upon reduction of the snow cover and the possible development of a bottom ice algal





bloom, are critical aspects when simulating ice algal phenology and attempting to quantify the contribution of sympagic algae to Arctic primary production.

Simulated turbulent diffusivities are up to four orders of magnitude higher than molecular diffusivities and the results presented herein emphasize their potential role in sea ice biogeochemistry. The number and intensity of Arctic winter storms has increased over the 1979–2016 period (Rinke et al., 2017; Graham et al., 2017) and the effect of more frequent and more intensive winter storms in the Atlantic Sector of the Arctic Ocean is a thinner, weaker, and younger snow-laden ice pack (Graham et al., 2019). Storms that occur late in the winter season, after a deep snowpack has accumulated, have the potential to promote ice growth by dynamically opening leads where new ice growth can take place. The young ice of the refrozen leads does not have time to accumulate a deep snow layer until the melting season, which could lead to light limitation of algal growth. All things considered, it can be expected that ongoing trends in the Arctic will lead to a release from light limitation in increasingly larger areas of the ice pack in late winter, which will lead to more likely nutrient limitation earlier in spring (e.g. Lannuzel et al. 2020). These effects will be further amplified under thinning of the snowpack as observed in western Arctic, and in the Beaufort and Chukchi seas, over the last decades (Webster et al., 2014). Therefore, properly parameterizing nutrient exchanges between the ice and the ocean in sea-ice biogeochemical models is of utmost importance to avoid overestimating nutrient limitation and thus underestimating sea ice algal primary production.

In existing sea-ice models there are "natural" differences between the way budgets for non-conservative tracers such as nutrients are closed compared to those of momentum, heat and salt, which are related to the biogeochemical sinks and sources (e.g., equation 18 in Vancoppenolle et al., 2010), but also some "inconsistencies", related with the way their transfers between the ocean and the ice are computed. Interestingly, some models (e.g., Jin et al., 2006, 2008 and Hunke et al., 2016) apply the diffusion equation to calculate exchanges across the bottom ice not only to dissolved tracers, but also to algal cells. This is to guarantee a mechanism of ice colonization by microalgae. However, the usage of the same coefficient for dissolved and particulate components creates significant uncertainty.

Molecular diffusion is a slow process compared with momentum and heat turbulent exchanges. This justifies the usage of diffusion coefficients which are much higher than molecular diffusivity, as in Jin et al. (2006), using a value of $1.0 \times 10-5$ m$^2$ s$^{-1}$, four orders of magnitude higher than the value indicated in Mann and Lazier (2005) – $1.5 \times 10-9$ m$^2$ s$^{-1}$ – or the parameterization of diffusivity as a function of friction velocity as in Mortenson et al. (2017). The approach proposed herein, formulating bottom-ice nutrient exchanges in a way that is consistent with momentum and heat exchanges, provides a physically sound, consistent, and easy to implement alternative.

## 5. Conclusions

Replacing molecular with turbulent diffusion at the ice-ocean interface in a sea-ice biogeochemical sub-model, leads to a reduction in nutrient limitation and a significant increase in ice algal net primary production and *Chl a* biomass accumulation in the bottom-ice layers, when production is nutrient limited. The results presented herein emphasize the potential role of



bottom-ice nutrient exchange processes, irrespective of brine dynamics and other physical-chemical processes, in delivering
nutrients to bottom-ice algal communities, and thus the importance of properly including them in sea-ice models. The relevance
of this becomes even more apparent considering ongoing changes in the Arctic icescape, with a predictable decrease in light
limitation as ice becomes thinner and more fractured, with an expected reduction in snow cover.
**Code availability**
The software code used in this study may be found at:
https://doi.org/10.5281/zenodo.4675097 and https://doi.org/10.5281/zenodo.4675021
This code is in a fork derived from the CICE Consortium repository (https://github.com/CICE-Consortium).
The Consortium's codes are open-source with a standard 3-clause BSD license and are is under the following Copyright
license, available at (https://cice-consortium-cice.readthedocs.io/en/master/intro/copyright.html):

**Data availability**
Model forcing function files may be found at: https://doi.org/10.5281/zenodo.4672176
Results from model simulations described above, in the form of CICE daily netCDF history files iceh.* may be found at:
http://doi.org/10.5281/zenodo.4672210
There is one directory for each simulation, and it includes besides the historical files the input file (ice_in) with the simulation
parameters.

**Authors contribution**
Pedro Duarte made the software changes, designed the experiments, performed the simulations and prepared the manuscript
with contributions from all co-authors.
Philipp Assmy contributed to the writing of the manuscript.
Karley Campbell contributed to the writing of the manuscript.
Arild Sundfjord contributed to the writing of the manuscript and to funding acquisition.

**Competing interests**
The authors declare that they have no conflict of interest.
**Acknowledgements**
This work has been supported by the Fram Centre Arctic Ocean flagship project "Mesoscale physical and biogeochemical
modelling of the ocean and sea-ice in the Arctic Ocean" (project reference 66200), the Norwegian Metacenter for




Computational Science application "NN9300K - Ecosystem modelling of the Arctic Ocean around Svalbard", the Norwegian "Nansen Legacy" project (no. 276730) and the European Union's Horizon 2020 research and innovation programme under grant agreement No 869154. Contributions by K Campbell are supported by the Diatom ARCTIC project (NE/R012849/1;03F0810A), part of the Changing Arctic Ocean program, jointly funded by the UKRI Natural Environment Research Council and the German Federal Ministry of Education and Research (BMBF).

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
