# Peer review of "The importance of turbulent ocean-sea ice nutrient exchanges for"

_Geoscientific Model Development, 2021_

## Referee Comment (RC1)

**Review of gmd-2021-58**

**The importance of turbulent ocean-sea ice nutrient exchanges for simulation of ice algal biomass and production with CICE6.1 and Icepack by Pedro Duarte, Philipp Assmy, Karley Campbell, and Arild Sundfjord**

M. Vichi, Department of Oceanography, Marine and Antarctic centre for Innovation

June 2021

**1 Introduction**

The main point of this manuscript is that most of sea-ice biogeochemistry models do not include a proper treatment of turbulent exchanges between the ocean and sea-ice brines. I commend the authors for raising this important issue and for testing the consequences on a dedicated numerical setup, but I would argue that their argumentation may lead to further misinterpretation. It is not a matter of neglecting the turbulent exchange at the sea-ice/ocean interface, but rather making a proper overall consideration of the underlying physical processes. Contrary to the authors' claim, this has indeed been taken into account in the published literature although the lack of direct explanations on the underlying assumptions may have contributed to misinterpretations. I think this is a very good manuscript that would deserve publication, since it is going to contribute to the development of more adequate parameterizations of sea-ice fluxes, especially in the likelihood of a nutrient-limited future Arctic Ocean, as pointed out by the authors in the introduction. I however think the manuscript would benefit from additional work on the current version to address the substantive concern detailed below.

**2 General comment**

The recommendation done by the authors is that nutrient exchanges (and by extension any material flux) at the bottom interface with sea ice should be consistent with the way heat and salt fluxes are parameterized. This is indeed a reasonable advice, which in my opinion it has been taken into account in the literature. There are some theoretical differences in the proposed approaches that need to be taken into consideration, and I would suggest the authors to include a slightly more extended explanation in their background analysis. It is true that a series of refinements have been added to the description of momentum, heat and salt exchanges in sea-ice dynamics. Models were initially considered isohaline, and only heat conduction was considered. The various simplifications were eventually reconsidered and expanded as new knowledge was available. I recognize

that a similar approach has not been taken for the nutrient and, more in general, material exchanges at the water interface. This led to a lack of synchronicity in the development of the physical and biogeochemical components.

The parametrization proposed by Cota et al. (1987) is basically a formulation of Fick's law of diffusion. This is valid everywhere it is possible to determine a diffusivity coefficient (molecular or turbulent). However, at the interface between ocean and sea ice, we deal with a combination of turbulent and advective flux due to the physical growth of sea ice.

Equation (2) from McPhee (2008) referenced by the authors is only one component of the full salt conservation equation at the interface (Fig. 6.3 and eq. 6.3 in McPhee, 2008, but written here as eq. 6.8 and using the same notation as used by the authors):

$$\alpha_S u^* \left(S_w - S_0\right) + w \left(S_i - S_0\right) = 0 \tag{1}$$

where $S_0$ is salinity at the ice-water interface, $S_i$ is the brine salinity and $S_w$ is water salinity in the far field (generally the mixed layer salinity)

Most of the earlier publications made the assumption (even if not explicitly stated) that salinity at the interface is equal to mixed-layer salinity, $S_0 = S_w$, and hence the first term vanishes and the second term becomes $w \left(S_i - S_w\right)$. The physical implication is that turbulent exchange is assumed to be quicker than any other process, the solute is vertically homogeneous in the mixed layer and hence salt flux is mainly regulated by the entrapment/release flux due to sea-ice growth

$$w = \frac{dh}{dt},$$

where $h$ is sea-ice thickness.

Turbulence is less likely to occur within or through the pores of the brine channels because they are microscopic, usually smaller than the turbulent eddies found in the vicinity of the interface. Molecular diffusion should prevail. But turbulence, expressed here by the friction velocity and the non-dimensional scaling coefficient $\alpha_S$ does affect the actual concentration at the interface ($S_0$). This is the the concentration that would then be entrapped in the sea ice, as well as diffused at low Reynolds numbers (usually discarded). The authors use this equation to prescribe a nutrient flux (their eq. 4):

$$F_N = -\alpha_S u^* \left(N_w - N_i\right),$$

which implies that the solute concentration at the interface (what should be $N_0$ in accordance with McPhee's formulation) is now equivalent to the brine concentration ($N_0 = N_i$). Hence, the advective flux is neglected and the flux is fully regulated by water column turbulence (as if the interface with sea ice would be equivalent to the interface between two layers of water). The change of sign with respect to the salt equation is however unclear and should be explained. If the system of reference is oriented upwards, this would lead to a negative flux when the water concentration is higher than the concentration in the brines, but I stand to be corrected.

To my knowledge, there are no dedicated measurements that demonstrate which approximation is superior, hence the claim that the published models are neglecting an important flux should be reformulated. Table 1 (together with the introduction and part of the discussion) conveys a misleading signal, as if the works that did not include a specific parameterization of turbulent diffusion, did not have any flux at all. They instead incorporated the turbulent flux in the advective component. I also think the authors should more adequately address the difference between molecular and eddy diffusivity, and the way it has been employed in the literature. Models that resolve the bottom nutrient exchange through molecular diffusion (and neglecting the advective component) are indeed more likely to underestimate this flux as shown in the results. I would also suggest the authors to clarify the argument of changing the timescales by adding some more explanation. This is pertinent to the CICE implementation of nutrient diffusion, while it is seems as a general approach used in the literature.

In summary, models should ideally resolve both terms of eq. (1) simultaneously, which is not possible without further assumptions, because there are two unknowns $(S_i, S_0)$ and one single equation (as opposed to the $T$,S system described in eq. 6.8 by McPhee, in which it is possible to thermodynamically connect the two variables).

Having said this, there is a lot of merit in the results shown in this manuscript. They show the role of assuming full permeability of the ice-water interface, as if the brines would be covering the whole surface and be affected by turbulence as a layer of water. In this context, the role of $\alpha_S$ and the related time scale becomes dominant, as clearly shown by the authors in the result section (They also state that this parameter is usually different when sea ice is growing or melting, which is another indication of the importance of advective processes and the co-existence of the two). The claim that the published models underestimate bottom sea-ice algae production because they do not resolve the turbulent fluxes is not substantiated by the presented analysis, although the authors are clearly showing that the parameterization choices lead to a different evolution of sea-ice algae biomass.

**3 Specific comments**

L58-59      I would suggest to report the units of $\Delta C$ and $\Delta z$ separately, and not the units of the ratio

L68-70      This sentence should be changed in light of the main comment above. It contributes to the lack of clarity that the authors are indeed trying to address. Diffusion and advection are two separate processes.

L88-89      The symbol $\alpha$ is not the same in the text and in eq. (2)

L105-122   It should be clarified that this implementation of the diffusion process and the difference in scales is due to the choices done in CICE

L148-151   I would suggest the authors to include a (very) brief description of the simulation set-up carried out in Duarte et al., (2017), especially in terms of how

the nutrient far-field is prescribed.

L153-155&179 Many other parameters were sequentially changed, and not always one at a time. It is thus difficult to appreciate the role of each one. I understand that one of the authors finding is that they had to artificially alter other parameters in order to supplement for the limited nutrient fluxes simulated by a molecular diffusion parameterization. I wonder if this could be presented in simpler terms without the many experiments shown in Table 2, which do not always contribute to the aim of this manuscript.

L213 The $\alpha_s$ values should be presented in the text and not just quickly in the caption, and further discussed if possible. This becomes a crucial parameter as highlighted in Sec. 2. (please use a space for scientific notation for all the numbers in the table, e.g. $8.6\,10^{-5}$)

L246-248 May I kindly request that the supplementary figures be prepared with experiments side by side as done in the manuscript? This would greatly aid the comparison.

L249 "CICE tracers" should probably be "CICE diagnostics"

L251 Figure 5 shows the direct consequence of the large change in diffusivity values. This figure does not appear to be fundamental and could be moved to the supplementary. A figure on the light limitation would instead be helpful, since this process is discussed in Sec. 4

L262 I cannot see the magenta line

L288 I think the authors mean "followed by silicate"

L292-293 This sentence is unclear and I struggled to interpret it. Is it the standing stock at the end of the ice period? They appear quite similar to me.

L299-301 This is also a direct consequence of the difference in magnitude. It is also not very visible. A comparison of the nutrient flux using the prescribed eq. (4) from the manuscript would have been more helpful

L306-307 This does not explain why light is less limited on June 1st in Sym2 with less snow with respect to Sym1. Please clarify

---

## Author Comment (AC1)

Dear Marcello Vichi,

Thank you very much for your constructive comments that we used to improve the manuscript. In the following text we will address your comments and explain how they were incorporated into the revised manuscript. We follow the structure of your document divided in **Introduction**, **General comment,** and **Specific comments**. We also considered the corrections you made in a second file.

The citations used below were also used in the paper. Therefore, you may find the corresponding references in the bibliography of the manuscript.

**1. Introduction**

Here you show your disagreement with the idea that most of sea-ice biogeochemistry models do not include a proper treatment of turbulent exchanges between the ocean and the sea-ice brines. Under the **General comment** you further develop on this opinion. Our understanding is that this is your main critique of our manuscript, and we will focus our responses mostly on this aspect.

It seems to us that part of your disagreement results from the lack of clarity of some of our statements. In fact, we felt the same after reading the comments from the second referee (Martin Vancoppnelolle). Therefore, we implemented several changes in the revised manuscript in order to try to make it clearer.

We would like to emphasize that we are focusing on turbulent exchanges between the ocean and the sea-ice that are independent of ice growth/melting and of brine drainage and that are driven by current velocity shear. Whereas sea-ice growth or melting should not be confounded at all with our focus, since they imply bulk exchanges, brine drainage may well be confounded with the turbulent exchanges emphasized in our manuscript, since it may also be associated with turbulence (e.g. Jeffery et al., 2011). Please refer the synthesis at the penultimate paragraph of the Introduction section that we reproduce here after changes introduced in the revised manuscript (the bold type is just to emphasize the processes that are our focus):

"From this assessment one may divide the ocean-ice exchange processes of existing biogeochemical models into those related to: (i) entrapment during freezing; (ii) flushing and release during melting;(iii) brine gravity drainage, driven by density instability, parameterized as either a diffusive or a convective process; (iv) molecular diffusion; (v) **turbulent diffusion at the interface between the ocean and the ice induced by velocity shear – the focus of this study**."

As a result, the processes listed here include but are not limited to those mentioned in your review, as our focus is rather on the exchanges taking place irrespective of brine drainage and of ice growth/melting. We tried to better emphasize this idea in the revised manuscript. In the abstract we also added some text to avoid the implication that other sea-ice biogeochemistry models do not include any exchange processes. We changed the sentence:

"We hypothesize that biogeochemical models which do not consider such turbulent nutrient exchanges between the ocean and the sea-ice underestimate bottom-ice algal production."

To:

"We hypothesize that biogeochemical models which do not consider such turbulent nutrient exchanges between the ocean and the sea-ice, **despite considering brine drainage and bulk exchanges through ice freezing/melting**, **may** underestimate bottom-ice algal production."

We made compatible changes in the penultimate paragraph of the Introduction section, that now reads as:

"We hypothesize that models which do not consider **the role of current velocity shear on turbulent nutrient exchanges between the ocean and the sea-ice** may underestimate bottom-ice algal production."

We also made other changes in the Introduction also related with comments from the other referee an in line with the need to clarify better our approach.

**2. General comment**

In the first paragraph, you seem to agree with one of our main points that **nutrient exchanges at the sea ice bottom interface should be consistent with the way heat and salt fluxes are parameterized**. In the next paragraphs you use McPhee's formulation (presented as equation 1) to analyze further the various fluxes involved in nutrient exchanges and the underlying physical processes. Please note that we focused only on the first term of McPhee's equation – the one correspondent to turbulent exchanges – whereas you focus on both terms. Please refer the last sentence in the first paragraph of the Introduction. We added a sentence to the last paragraph of the Introduction section to emphasize this focus:

"To test the above hypothesis, we use a 1D vertically resolved model and contrast results using the default diffusion parameterization and a "turbulent" parameterization analogous to that of momentum and heat transfer, at the interface between the ocean and the sea ice, based on McPhee (2008)."

After presenting McPhee's equation, you write that most of the earlier publications assumed that interface and far-field salinities are equal and, therefore, the term which is the focus of our study vanishes. If you mean "model publications" we argue that, in fact, several models (listed in Table 1) consider diffusion exchanges based on concentration differences between the ocean and the bottom ice (it seems that for practical reasons bottom ice concentrations are used as a proxy for interface concentration). The point we emphasize in our study is that by doing so, they do not treat it as a turbulent process consistently with the calculation of momentum or heat exchanges. This aspect is emphasized in the manuscript. Please refer to the first two lines of the Abstract and the penultimate paragraph of the Introduction. As a side note we emphasize several corrections made to Table 1 regarding the diffusion approaches used in various models.

When discussing the second term of McPhee's equation you imply that when the first term is discarded, salt and, presumably, nutrient fluxes become mostly regulated by entrapment/release due to ice growth/melting, gravity drainage and percolation (these last two were added in your Correction comments). Whatever processes are included here, they depend on $w \neq 0$, i.e., ice must be either growing or melting or there must be some brine drainage. The processes we focus on, occur irrespective of ice growth or melting or brine drainage. So, if ice thickness is not changing and brine is not moving, you still have bottom exchanges of nutrients as long as gradients exist between the bottom ice and the water. It is the role of these and only these fluxes we are addressing here. Please refer the last paragraph of the Introduction.

Regarding the comment on the minus sign used in manuscript equation 4, we added an explanation in the revised manuscript, immediately after presenting the equation - the minus sign follows the CICE and Icepack convention of considering negative the upward fluxes.

You noted the apparent absence of dedicated measurements to demonstrate which approach is superior. We are not aware of such measurements either. However, the studies by Cota et al. (1987) and by Dalman et al. (2019) (cited in our manuscript) provide evidence for the possible importance of turbulent exchanges induced by current velocity shear, that are the focus of our study, in supplying nutrients to the bottom ice and stimulating algal growth. In fact, we are planning to experimentally address the bottom ice nutrient diffusion processes within the scope of a recently approved Research Council of Norway project called BREATH, under the lead of Karley Campbell (one of the co-authors of this manuscript).

You mention that we should more adequately address the difference between molecular and eddy diffusivity, and the way it has been employed in the literature. Perhaps we misunderstood you comment but it seems to us this is already done in lines 51-66 of the preprinted manuscript and also in Table 1. However, we also added a new paragraph to the end of section 2.1 about molecular and turbulent diffusion and comparing their magnitudes. Moreover, in the revised manuscript we refer to other forms of brine exchanges that also may imply turbulent diffusion but that are not induced by velocity shear – refer the antepenultimate and penultimate paragraphs of the Introduction.

You suggested that we should clarify the argument of changing the timescales by adding some more explanation. As we understood you meant that we should explain that what we did is pertinent to the CICE implementation of nutrient diffusion but may not be applicable to other models. In line with this interpretation, we added some text at the end of section 2.1 to emphasize the specificity of this "time scale approach" to the CICE model. Moreover, and in line with comments from the other referee, we added the transport equation to 2.1 and detailed the part of the equation that we changed.

Regarding your comment about our claim that the published models underestimate bottom sea-ice algae production because they do not resolve the turbulent fluxes induced by velocity shear, we merely found evidence supporting such hypothesis. But, as you noted and we commented above, we lack experimental evidence to support our conclusions, despite the works of Cota et al. (1987) and Dalman et al. (2019), suggesting that turbulence may indeed enhance bottom ice algal growth by increasing nutrient fluxes. Therefore, in the revised manuscript version we limited the perceived certainty of such claims. In the abstract we replaced "underestimate" by "may underestimate" and we added a sentence emphasizing the need for experimental evidence. Moreover, we added two sentences to the first paragraph of the Discussion that now became:

"The results obtained in this study support the initial hypothesis, showing that replacing molecular with turbulent diffusion at the interface between the ocean and the sea ice, formulated in a way consistent with momentum and heat exchanges, leads to a reduction in nutrient limitation that supports a significant increase in ice algal net primary production and Chl a biomass accumulation in the bottom ice layers, when production is understood to be nutrient limited. Therefore, our results are in line with empirical evidence provided by Cota et al. (1987) and Dalman et al. (2019) but, to the best of our knowledge, experimental evidence from properly dedicated experiments is still lacking to test our hypothesis. Moreover, our results do not imply necessarily that experiments carried out with other sea-ice models would render the same trends."

In line with these changes we also did some changes in the first sentence of the Conclusions which now became:

"Considering the role of velocity shear on turbulent nutrient exchanges at the interface between the ocean and the ice in a sea-ice biogeochemical sub-model, leads to a reduction in nutrient limitation and a significant increase in ice algal net…"

**3. Specific comments (referee comments in italics)**

*L58-59 I would suggest to report the units of $\Delta C$ and $\Delta z$ separately, and not the units of the ratio*

**Answer:** Done as suggested.

*L68-70 This sentence should be changed in light of the main comment above. It contributes to the lack of clarity that the authors are indeed trying to address. Diffusion and advection are two separate processes.*

**Answer:**
The sentences mentioned by the referee were reproduced above after rewriting for the revised manuscript. We hope that the changes done in this part of the manuscript (penultimate paragraph of the Introduction) make the whole meaning of our sentences clearer.

*L88-89 The symbol $\alpha$ is not the same in the text and in eq. (2)*

**Answer:** Corrected.

*L105-122 It should be clarified that this implementation of the diffusion process and the difference in scales is due to the choices done in CICE*

**Answer:** Done as suggested. Please see the changes we did in 2.1.

*L148-151 I would suggest the authors to include a (very) brief description of the simulation set-up carried out in Duarte et al., (2017), especially in terms of how the nutrient far-field is prescribed.*

**Answer:** Please note that in the first sentences under "2.3 Model simulations" we provide a general description of the model setup. At the last paragraph of this section, we give details on model forcing, including nutrient forcing. Following your advice, we added some words to this paragraph, specifying how water column forcing was considered in the model, namely, that ocean forcing is based on measurements carried out within the surface 2 meters. In the revised manuscript this last paragraph became the first one, following some critics from the other referee regarding the organization of this section.

*L153-155&179 Many other parameters were sequentially changed, and not always one at a time. It is thus difficult to appreciate the role of each one. I understand that one of the authors finding is that they had to artificially alter other parameters in order to supplement for the limited nutrient fluxes simulated by a molecular diffusion parameterization. I wonder if this could be presented in simpler terms without the many experiments shown in Table 2, which do not always contribute to the aim of this manuscript.*

**Answer:** We did considerable changes in Table 2 to make it more readable. The relatively large number of simulations spanning two ice types and considering several parameter changes are not easy to follow. We hope these changes in Table 2 make it easier to understand the logic behind the various modeling experiments.

*L213*
*The $\alpha$s values should be presented in the text and not just quickly in the caption, and further discussed if possible. This becomes a crucial parameter as highlighted in Sec. 2. (please use a space for scientific notation for all the numbers in the table, e.g. 8:6 $10^{-5}$).*

**Answer:** We added the $\alpha$s values to the text in section 2.1, 3rd paragraph. We also corrected all "powers" that had a multiplication sign.

*L246-248 May I kindly request that the supplementary figures be prepared with experiments side by side as done in the manuscript? This would greatly aid the comparison.*

**Answer:** The organization of the figures in the manuscript and in supplementary info follows the same sequence and organization so it is not clear what is meant regarding the preparation of the figures with the experiments side by side. Figures either include experiments 1, 2, 3, 4 and 5 or experiments 6, 7, 8 and 9, in the case of the Refrozen lead and both in the paper and in Supplementary info. Sequences and organization for second year ice simulations are similar.

*L249 "CICE tracers" should probably be "CICE diagnostics"*

**Answer:** Corrected as suggested.

*L251 Figure 5 shows the direct consequence of the large change in diffusivity values. This figure does not appear to be fundamental and could be moved to the supplementary. A figure on the light limitation would instead be helpful, since this process is discussed in Sec. 4.*

**Answer:** We prefer keeping the present organization. Please note that we show in Figure 3 the main limiting factor results for the refrozen lead, which is Si. We also would like to keep Figure 5 in the paper because it shows results directly reflecting the usage of different diffusion approaches/parameters, which is one of the main topics of this study.

*L262 I cannot see the magenta line*

**Answer:** Parts of the magenta line are visible but most of it is under the green line. We added some text to the figure captions explaining this.

*L288 I think the authors mean "followed by silicate".*

**Answer:** Corrected as suggested.

*L292-293 This sentence is unclear and I struggled to interpret it. Is it the standing stock at the end of the ice period? They appear quite similar to me.*

**Answer:** The sentences read as:

"Maximum Chl a values predicted for SYI are between two and three orders of magnitude lower than those predicted for the RL (Figs. 2 and 7). However, standing stocks for the former are larger than those for the latter, considering both observational and model data (Figs. 1b and 6)."

We changed this sentence to:

"Maximal Chl a concentration predicted for the RL_Sim1 and RL_Sim5 simulations - those closer to observations - are two orders of magnitude higher than those predicted for SYI (Fig. 2a and e *versus* Fig. 7). However, standing stocks predicted for RL_Sim1 and RL_Sim5 simulations are smaller than for SYI simulations, as confirmed by the observations (Figs. 1b and 6)."

*L299-301 This is also a direct consequence of the difference in magnitude. It is also not very visible. A comparison of the nutrient flux using the prescribed eq. (4) from the manuscript would have been more helpful.*

**Answer:** Please note that we are showing only diffusivity to emphasize differences that are only due to the used formulation (refer 2.1 and the many changes done in this section following suggestions from you and the second referee). If we plotted instead nutrient fluxes, differences between both graphs would have resulted also from the nutrient gradients in both simulations.

*L306-307 This does not explain why light is less limited on June 1st in Sym2 with less snow with respect to Sym1. Please clarify.*

**Answer:** In fact, light is more limiting in SYI Sim2 on June $1^{st}$, especially for the simulation with less snow (limiting light values lower => more limitation). The reason for this is the higher chlorophyll concentration in SYI Sim2 (please compare Fig. 11g and h), resulting from less Si limitation, and blocking light more efficiently – in the CICE model sea ice optical properties are influenced by chlorophyll concentration.

---

## Author Comment (AC4)

Dear Martin Vancoppenolle,

Thank you very much for your constructive comments that we used to improve the manuscript. In the following text we will address your comments and explain how they were incorporated in the revised manuscript. We follow the structure of your document with some general introductory remarks followed by **General comment** and **Specific comments**.

The citations used below were also used in the paper. Therefore, you may find the corresponding references in the bibliography of the manuscript.

In your introductory remarks you write:
"Whereas I believe the authors make a key, nice and novel point, the means they currently use are not robust enough. One key problem is that the processes below and above the ice-ocean interface are not clearly distinguished and treated together."

We believe that this results from a lack of clarity in our previous manuscript version. Since you develop further on this topic along your comments, we will try to clarify better our approach in this response and in the revised manuscript.

**General comment**

**Your first critique** concerns the lack of clarity about whether we act on the nutrient flux below the ice/at the top of the mixed layer or within the ice. Whilst the model setup we used considers both nutrient fluxes **within** the ice and at the **interface** between the mixed layer and the bottom ice (in the words of McPhee "salt balance at the **interface**"), our experiments acted only upon the **interface** nutrient fluxes. Please refer to the first paragraph of 2.1 Concepts, where we did some changes in the revised manuscript to make the whole ideas clearer:

" Eq. (1) from Cota et al. (1987) provides the basis for our reasoning about nutrient exchanges between the ocean and the sea-ice bottom being based on a turbulent exchange process enhanced by current velocity shear, irrespective of other exchanges based on brine dynamics, ice melt and ice growth. These turbulent exchanges may be parameterized through the flux of a quantity at the interface between the ocean and the sea ice, calculated as the product of a scale velocity and the change in the quantity from the boundary to some reference level (McPhee, 2008):"

Please note that we assume that nutrient exchanges at the interface are comparable to salt fluxes and governed by the same physics. Given the physical discretization of the biogrid (the biogeochemical model grid used in CICE), these interface fluxes directly affect the properties of the ice bottom layer only. These effects "propagate" vertically through diffusion within the ice or brine exchanges.

We did some changes to the caption of Table 1 to specify better our focus. Now it reads as:

"**Table 1.** Model parameterizations used/proposed by different authors to compute diffusion of nutrients. The only example based on friction velocity is that of Mortenson et al. (2017). "None" is used when exchange processes depend solely on ice growth/melting**."**

As you may see in Table 1 contents and in the changes made elsewhere in the text, now we clearly acknowledge that brine dynamics, as treated in some models, may be seen as a turbulent diffusion process. Please see also the antepenultimate paragraph of the Introduction that now reads as:

"Table 1 summarizes several models published over the last decades and their approaches to the calculation of tracer diffusion. Some models do not consider this process or limit it to molecular diffusion. Other models consider turbulent exchanges parameterized as a function of the Rayleigh number, calculated from brine vertical density gradients. Only one of the sampled models (Mortenson et al., 2017) uses a parameterization based on friction velocity."

Please see also the penultimate paragraph of the Introduction:
"From this assessment one may divide the ocean-ice exchange processes of existing biogeochemical models into those related to: (i) entrapment during freezing; (ii) flushing and release during melting;(iii) brine gravity drainage, driven by density instability, parameterized as either a diffusive or a convective process; (iv) molecular diffusion; (v) turbulent diffusion at the interface between the ocean and the ice induced by velocity shear – the focus of this study. In the absence of ice growth and when brine gravity drainage is limited, diffusive nutrient exchanges between the ocean and the ice have the capacity to limit primary production…"

We did a clear distinction between the turbulent diffusion driven by velocity shear, which is the focus of our study, and other forms such as molecular diffusion or diffusion driven by hydrostatic instability.

**Your second critique** is about the experimental setup, and you criticize the lack of explanations about which diffusion scheme we use. You also cite Jeffery et al. (2010) about the two existing diffusion schemes. We are not aware of the paper you cited but we cite a paper from 2011 (Jeffery, N., Hunke, E. C., and Elliott, S. M.: Modeling the transport of passive tracers in sea ice, J. Geophys. Res.-Oceans, 116, Artn C07020, doi:10.1029/2010jc006527, 2011) where two diffusion schemes are described. We use the Mixed Length Diffusion scheme described in this paper. This is now explained in the revised manuscript. Please refer 2.1 and 2.3.

We agree that here the reader may become unclear about the details of our settings, and the separation between within ice and interface processes, in line with your concerns. Therefore, we reordered the topics presented in 2.3 and, in the third paragraph, where we describe model settings, we write:

"We ran simulations with the standard formulations for biogeochemical processes described in Jeffery et al. (2016) and settings described in Duarte et al. (2017), using mushy thermodynamics, vertically resolved biogeochemistry, and including: freezing, flushing, brine molecular and mixed length diffusion within the ice and at the interface between the ocean and the sea ice as nutrient exchange mechanisms (Jeffery et al., 2011, 2016)...We contrasted the above simulations against others that replaced brine molecular and mixed length diffusion of nutrients at the interface between the ocean and the sea ice with diffusion driven by current velocity shear (Table 2), calculated similar to heat and momentum exchanges, and following the parameterization described in McPhee et al. (2008) and detailed above (equations 2 – 10)"

Please note also major changes in 2.1.

You also wrote "The limitation factors could be described with a sentence or two and values of the key parameters could be given, as the results refer to those extensively".

In the revised manuscript, we detail the values of the parameters used in the calculation of diffusion driven by velocity shear. We also add more details about Mixed Length Diffusion, but this is merely repeating info that is given in Jeffery et al. (2011). Therefore, we avoided many details and cited the mentioned authors for more info. Please see 2.1.

**Your third critique** is about a better description of the state of the art and more connection between our results and the existing literature. We will address below each of the points you raised.

Your first point is about the existing forms of enhanced diffusion described in papers that are already cited in our manuscript (we repeat here that we are not aware of the paper you cite "Jeffery et al. (2010)" but we cite Jeffery et al. (2011) which describes enhanced molecular diffusion and mixing length diffusion as parameterizations of gravity drainage). Please refer Introduction, 2nd paragraph:

"More recently, other authors have integrated formulations based on hydrostatic instability of brine density profiles, to compute brine gravity drainage and tracer exchange between the ice and sea water, based on diffusive (Vancoppenolle et al., 2010; Jeffery et al., 2011)…"

We changed this sentence to specify "enhanced diffusion" and make it clearer what was done by the cited authors:

"More recently, other authors have integrated formulations of "enhanced diffusion" (Vancoppenolle et al., 2010; Jeffery et al., 2011) or convection (Turner et al., 2013), based on hydrostatic instability of brine density profiles, to compute brine gravity drainage and tracer exchange within the ice and between the ice and the sea water. Comparisons between salt dynamics in growing sea ice with salinity measurements showed that convective Rayleigh number-based parameterizations (e.g. Wells et al., 2011), such as the one by Turner et al. (2013), outperform diffusive and simple convective formulations (Thomas et al., 2020)."

In this first point you also mention **"Still the present paper is original, because of the focus on the melt period"**. In fact, we are not focusing only on the melting period. Please note that we write in the penultimate paragraph of the Introduction:

"In the absence of ice growth and when brine gravity drainage is limited, diffusive nutrient exchanges between the ocean and the ice have the capacity to limit primary production …"

Moreover, our simulations also cover periods of ice growth, in the case of the refrozen lead, and periods when ice thickness was quite stable, especially in the case of second year ice.

We hope that the changes made to the last paragraph of the Introduction will help clarifying better the focus of this study.

Your second point is about the experimental support for advective approaches for brine convection. Thank you very much for letting us know about the study by Thomas et al. (2020), providing evidence for the accuracy of Rayleigh number-based parameterizations for predicting sea ice bulk salinity. We added some text to the paper specifying these findings by Thomas et al. (2020). A last sentence was added to the second paragraph of the Introduction:

"Comparisons between salt dynamics in growing sea ice with salinity measurements showed that convective Rayleigh number-based parameterizations (e.g. Wells et al., 2011), such as the one by Turner et al. (2013), outperform diffusive and simple convective formulations (Thomas et al., 2020)."

Your third point is about the description of physical processes, suggesting there is room for improvement and emphasizing two aspects: relative ice-ocean velocity and forced convection of brine. In the revised manuscript, we added a new paragraph at the end of 2.1 (partly following critics from the other reviewer - Marcello Vichi) where we specify better some of the relevant physical processes. Here we also refer to **relative ice-ocean velocity** when talking about "stream" velocities. Moreover, we added a sentence about the work by Dalman et al. (2019) and a sentence about **forced convection** at the end of this new paragraph and following your suggestion.

In the fourth point you suggest that what is original in our work is the introduction of the extra nutrient source when ice is not growing. In fact, such an extra source exists also in models that consider molecular diffusion at sea-ice interface. The main point of our study is to evaluate the effect of velocity shear in diffusion in line with "forced convection", for example. Moreover, this nutrient source is acting always. You write that you "suspect that NPP in the different experiments split exactly when the temperature gradient within the ice reverses, which switches off gravity drainage (brine convection)". In fact, looking into Figure 1b (dashed lines), you may see that NPP splitting occurs before middle May in the refrozen lead simulations and that was when ice was still growing, and brine being produced. But you are absolutely right when you mention the increase in the production period. We added a few words about this to the second paragraph of the Discussion.

In the fifth point you criticize the quality of some figures and suggest renaming the simulation experiments. Regarding figure quality, we believe that the problem is only with the figures included in the pdf version due to their low resolution. We will provide high resolution figures at 300 dpi. Concerning simulation naming, we prefer keeping them as they are for the sake of simplicity. If we add some suffix linked to the type of the simulation as you suggest, and considering that RL_Sim6 – 9 repeat RL_Sim1-4, except for the starting date, we would have to had a number or something else to distinguish the former from the latter simulations and we prefer to keep naming as simple as possible – this becomes more practical when it comes to insert legends within the figures, for example. We hope that the considerable reworking of Table 2 will help the reader tracking the various simulation experiments.

**Specific comments (referee comments in italiscs)**

*l. 28. I would use « released » not « published »*

**Answer:** Done as suggested.

*l. 30 I would use « exchange » not « diffusion » of tracers, since there can also be advection of nutrients at the ice base*

**Answer:** Yes, but here we are explaining the focus of our work which is on the diffusion and not in the exchange in general.

*l. 34 I think here the references of Vancoppenolle et al QSR 2013 and more probably Thomas et al (The cryosphere 2020) would be appropriate, as they review such approaches.*

**Answer:** Done as suggested.

*l. 35 I would use « brine transport » instead of diffusive and convective fluxes*

**Answer:** Done as suggested.

*l. 33-42 Here the reference to Notz and Worster (JGR 2009) might be worth to consult and invoke, as they provide the current state-of-the art for brine dynamics. For gravity drainage, Wells et al (GRL 2011) would probably be best in terms of physical understanding.*

**Answer:** We added a reference to Wells in the lines you suggested, when writing about Rayleigh formulations. In these lines we focus on processes used to model nutrient exchanges. However, in the following paragraph we mention desalinization and the main processes associated with it. Therefore, following one of your suggestions we added here a citation to Notz and Worster that was missing.

*l. 43-49 Here I think the writing reflects some confusion between processes in the ice and below. In the cited papers, Vancoppenolle and Turner use an infinite salt / heat reservoir assumption (constant mixed layer salinity/temperature) and therefore do not need to specify what happens in the water column and use McPhee formulas.*

**Answer:** It seems to us that irrespective of considering or not what is happening in the water column and even assuming it as a constant and infinite reservoir, changes in the ice brine nutrients, as a result of biogeochemical processes, imply gradients at the sea-ice interface. These gradients should drive some exchanges.

*l. 43-49 In a 1D context, I would not refer to momentum transfer (it is not very useful since in such a context).*

**Answer:** You are right in that the focus of the paper is not on momentum transfer. However, momentum, heat and salt or nutrient transfers occur altogether at the sea-ice interface and that is why we would prefer keeping the emphasis on this communality.

*l. 57 use subscripts for « Fc » and Kz. Use SI units throughout the paper (m2/s).*

**Answer:** Done as suggested.

*l. 49 if you refer to âˆ†C, then units should be mmol/m3, if you refer to the product, then ok (but then writing needs to be corrected).*

**Answer:** This was corrected, also following a suggestion made by the other referee. Now we present separately the units, instead of the "product units" as before.

*l. 60 I think the reason why it has not been used is because many authors have modelled sea ice only, and not the under-ice water reservoir of nutrients (except possibly Tedesco and Vichi).*

**Answer:** You may well be right but, it seems to us that the relevance of the process does not depend on the way you consider the under-ice reservoir because changes in the sea ice will create gradients at some point, irrespective of the constancy of properties under the ice, and these gradients should drive fluxes.

*l. 66-77. Here I think you should more precisely describe the ice-ocean interface between what happens below (shear/buoyancy-induced mixing, cfr. McPhee) and above the interface (brine circulation, cfr Jeffery 2010, Vancoppenolle 2010, Thomas et al 2021).*

**Answer:** We changed this paragraph in the revised manuscript according to your comments.

*l. 66-68. « Brine drainage » as you refer to it should read « gravity drainage ». Flushing is understood as a brine drainage mechanism, so this should be reworded. I'd recommend to invoke Notz and Worster JGR2009 or Vancoppenolle et al QSR2013, which provide reviews on salt and nutrient transport physics, in order to be more precise in terms of wording.*

**Answer:** Corrected as suggested. The mentioned authors are cited before these lines when talking about each of these processes in more detail.

*Table 1 is misleading I think because what is meant by diffusion is ambiguous. For instance, Vancoppenolle et al 2010 use a diffusion equation within the ice. Also, the CICE model uses a diffusion that is not only molecular in the ice (see Jeffery et al 2011) whereas the table suggests it does. I would refer to « ocean-ice nutrient exchanges » and split between what happens below / within the ice. Below the ice, you could separate between the models which include some ocean reservoir, and the others which do not and assume infinite ocean reservoir. Next, you could possibly specify what models do within the sea ice. I would also remove the title of the paper in the column (« associated model»), which I found mismatching and not very helpful. You could also remove the table, I'm not sure it is useful.*

**Answer:** Please note that the caption to this table was changed in the revised manuscript. We hope that now it is clear that we are only focusing on diffusive processes taking place at the interface between the ocean and the sea ice.

**Your specific comments about Section 2 Methods**

**Answer:**

We did several changes in Sections 2.1 and 2.3 which we hope are in line with many of the aspects mentioned in your comments.

We removed the dimensional equations as suggested.

The separation between within ice processes and those at the interface was addressed before following your comments. Here we focus only on interface diffusion.

We commented before about the usage of diffusion parameterizations in CICE to model brine gravity drainage.

The nutrient boundary condition at the ice-ocean interface is the concentration at the upper mixed layer and this is now specified in the first paragraph of 2.3.

We did not remove references to heat in 2.2 because doing so would make it difficult to explain the relationship between the transfer coefficients for heat and for salt/nutrients and thereafter justify the values used in our study.

We corrected the subscripts and defined all symbols as suggested.

*L. 111-112 why multiplying D by porosity and what are these matrix coefficients ? I think ambiguities would be relieved if you gave the diffusion equation.*

**Answer**: We removed the sentence about the multiplication by porosity and we added many details and some equations to this section, including the diffusion equations.

**Your specific comments about Section 2.2**

*Section 2.2. I think it is nice to provide an implementation section, but this one looked a bit detailed for a scientific paper, especially because in comparison the physical / numerical implementation is not enough detailed.*

**Answer:** Please note that the numerical implementation is detailed in Jeffery et al. (2016), cited now in 2.1. We detailed the implementation for the sake of transparency and to make it easier for other users to reproduce what we did. However, if necessary, we may simplify this section.

*Setting a minimum value for u\* corresponds to assumptions on the relative ice-ocean current (you are assuming ice moves with respect to seawater, and it might help to acknowledge that).*

**Answer:** We mentioned relative ice-ocean velocity in the new (last) paragraph added to 2.1

*l. 138-141 I felt this a bit pointless in the context of the paper.*

**Answer:** These are practicalities about how we implemented our changes in CICE and Icepack. Without giving them, it might be difficult for someone else to reproduce our simulations. However, as stated above we may reduce the detail here.

**Your specific comments about Section 2.3**

*Overall the section would read better if field experiments, forcing, initial conditions and sensitivity experiments were better separated.*

**Answer:** We did quite some changes in this section in line with your concerns.

*I would also suggest to work on more talkative simulation names (see generic comments), and work in parallel for the two sites.*

**Answer:** We answered above to a similar comment.

*Table 2 is exhaustive but was quite painful to read. You can gain in communication efficiency by making it more compact, and simpler, and keeping the details elsewhere in the text.*

**Answer:** Please note that we changed radically Table 2.

*l. 152 What do you mean by brine freezing? What do you mean by molecular diffusion?*

**Answer:** We did not mean to imply "brine freezing" and we rephrased the sentence in the revised manuscript. We also specified "within the ice and at the interface between the ocean and the sea ice as nutrient exchange mechanisms" which is accounted for in CICE (refer Jeffery et al. (2011 and 2016) listed in the bibliography).

**Section 3**

*l. 236. « Top of the brine network ». Where is that?*

**Answer:** We rephrased the sentence to (1$^{st}$ paragraph of the Results section):

"Concentrations in the layers located between the bottom and the top of the biogrid, defined by the vertical extent (brine height) of the brine network (green lines in the map plots) (Jeffery et al., 2011)"

We also add more explanations about the biogrid in 2.1

*l. 241. Higher limitation can mean both things (use stronger limitation or higher limitation factor?).*

**Answer:** Replaced with "stronger" as suggested.

*l. 299. If you refer to such a thing as « interface diffusivity », you should clearly define what is meant there. Also, I would rather look at the nutrient flux at the interface, than at the diffusivity.*

**Answer:** We changed the last paragraph of 3.1 to:

"Interface diffusivity (one of CICE diagnostic variables, corresponding to the diffusion coefficient between adjacent biogeochemical layers and between the bottom layers and the ocean) for simulations with turbulent exchanges ($\alpha_s u^*h$) are up to two orders of magnitude higher at the bottom (diffusivity between the bottom layer and the ocean) than for control simulations with only molecular diffusion ($D_m$) or $D_m$ + the mixed length diffusion coefficient ($D_{MLD}$) (refer 2.1 and Fig. 5)."

We also changed the legend of Figure 5 to:

"**Figure 5.** Daily averaged results for the refrozen lead (RL) simulations 1-5: Simulated evolution of interface diffusivity as a function of time and depth in the ice (note the colour scale differences between the various panels). In (a) interface diffusivity corresponds only to the molecular diffusion coefficient ($D_m$) or to $D_m$ + the mixed length diffusion coefficient ($D_{MLD}$). In the remaining panels and at the bottom layer it corresponds to the turbulent diffusion coefficient ($\alpha_s u^*h$) (refer 2.1). Ice thickness is given by the distance between the upper and the lower limits of the maps. The upper regions of the graphs, above the green line with zero values, are above the CICE biogrid and have no brine network. The magenta line, partly covered by the green line, represents sea level. Refer to Table 2 for details about model simulations."

Please note that we are showing only diffusivity to emphasize differences that are only due to the used formulation (refer 2.1). If we plotted instead nutrient fluxes, differences between both graphs would have resulted also from the nutrient gradients in the different simulations.

**Section 4.**

*« replacing molecular with turbulent diffusion ». I'm not sure this is the correct wording for what has been done (see general comment).*

**Answer:** This sentence was now changed to:

"The results obtained in this study support the initial hypothesis, showing that considering the role of velocity shear on turbulent nutrient exchanges between the ocean and the sea ice, formulated in a way consistent with momentum and heat exchanges…"

These changes were done for consistency with the way our hypothesis was reformulated in the revised manuscript to avoid confounding different forms of diffusion.

*Regarding silicate half-saturation, there are papers in the Antarctic that have found the same thing (Lim et al., JGR 2019).*

**Answer:** This study is cited in the revised manuscript. Please refer 4[th] paragraph of the Discussion section.

*l. 367. Delta-Eddington parameter -> insufficient detail of what is meant here.*

**Answer:** We explain what this parameter represents, and we refer sources where more details can be found (Urrego-Blanco et al., 2016; Duarte et al., 2017). In the revised manuscript we also specified that the model positive bias in June mentioned in the 6[th] paragraph of the Discussion is a shortwave bias.

---

## Referee Report (RR1)

**Review of gmd-2021-58-revision**

**The importance of turbulent ocean-sea ice nutrient exchanges for simulation of ice algal biomass and production with CICE6.1 and Icepack by Pedro Duarte, Philipp Assmy, Karley Campbell, and Arild Sundfjord**

M. Vichi, Department of Oceanography,

Marine and Antarctic centre for Innovation and Sustainability,

University of Cape Town

August 2021

**1 General comment**

I would like to thank the authors for the extended and comprehensive answers to my comments. Many of my remarks have been addressed, and the current version is more clear in explaining that this paper is meant to explore the role of turbulent fluxes in the overall exchange of nutrients at the sea ice-ocean interface. I think this can be further ameliorated, especially in reducing the confronting tone used when referring to the previous literature. I have made a few comments in this regard at the end.

The arguments made in this paper are specifically oriented to and implemented in the CiCE+Icepack model. However, they are presented in a more conceptual way (Sec. 2.1 title is indeed "Concepts), which is meant to be rigorous and understandable by all scientists and not just by those familiar with the model. I commend this approach because there is a need for clarity in this regard, but I am not yet fully convinced that this revision addresses my concerns. I am actually more confused than before in what kind of turbulent flux the authors are addressing. My initial understanding was that the authors parameterised the turbulent flux at the interface with the seawater side. This ambiguity has also been expressed by Dr Vancoppenolle, who requested to add the description of the nutrient transport within the sea ice (he thus thought the diffusion within the ice was somehow addressed in the paper). It is clear that both reviewers interpreted the work differently. I'm afraid the submitted revision still requires some work to achieve an adequate degree of clarity.

Let's start from the turbulent flux (eq. 2 in the revision), which I understand now it is meant to address only one component of the total salt flux at the interface (the Reynolds decomposition):

$$\langle w'S' \rangle \simeq \alpha_S u^* \left( S_w - S_0 \right) \tag{1}$$

The left hand side represents the averaged co-variance of the turbulent fluctuations, and not the the interface "vertical velocity and salinity" as erroneously written in the revised manuscript. The right hand side is the approximated form of the turbulent flux through the use of a boundary layer "friction velocity", where, as now properly indicated by the authors, $S_0$ is salinity within this ice-water interface, and $S_w$ is water salinity in the far field (generally mixed layer salinity). The authors state that this method can be applied to the nutrient flux at the same interface. Note that this is an extension of the concept, not in agreement with McPhee, 2008 as stated at line 106, since McPhee does not make this argument for nutrients. Also, there is no mentioning in McPhee (2008) that the nutrient concentration at the interface (which should be $N_0$, in accordance with McPhee's notation) is now equivalent to the nutrient concentration in the brines ($N_0 = N_i$) inside the sea ice:

$$F_N = -\alpha_S u^* \left( N_w - N_i \right),$$

(thanks for clarifying the sign). This would require some further explanation; but here comes my (new) confusion. From line 110 onward, the description of the concept is focused on the interface **within** the sea ice, the biologically active layer where brines are connected. This is what distance $h$ in eq. (5) is said to be. I am very puzzled as to how the friction velocity can be applied within the brines region. To my knowledge this timescale indicates replenishment at the interface with the water, where the the nutrient concentration is $N_0$, and the distance should be related to that turbulent boundary layer represented through the friction velocity. I am lost if the authors state that this distance is a sea-ice layer! I did ask the authors to clarify this timescale in my earlier review, but this was not specifically addressed (unless I missed it). The timescale indicated in eq. (6-7) is an expression of the turbulence within the brines, and hence it should be made very clear what the authors mean when they say "comparable" (line L113). I agree with the argument of comparing the timescales to understand diffusion through the interface, but this cannot be made using the brine height as a reference distance in both timescales since the processes are occurring at the two different sides of the same interface.

What I find most confusing is the fact that eq (8) is valid within the brines, and the authors decided to change the diffusive term in a quite disputable way (that is, using a friction velocity that is only defined at the ocean water boundary). Maybe some clarity would come if the author would explain the relationship between $N$ and $N_i$ used in their equation (4). Nowhere in the text it is stated that this is only applied at the boundary (I see the vertical gradients in eq. 9). There may be an argument in doing this, which is by prescribing the turbulent flux at the boundary. In that case, it would be plausible to equate the Reynolds terms in the brines with the ones in the ocean, because that turbulence-driven flux is the same (this is why they are dimensionally the same quantity). But I do not see this done in the revised text.

I would thus disagree with the statement done at lines 132-134. I am therefore concerned that some of the changes reported by the authors may be due to inappropriate handling of the equations, which would lead to a code implementation that is not justified by the underlying mathematical concepts. If the authors have changed eq (8) with

eq. (9) in the computation of the sea ice nutrient concentration within the sea ice, they have thus assumed that turbulent diffusivity **within the brines** (and not just at the boundary) is identical to the one in the ocean boundary layer. I am not surprised that the resulting nutrient concentration in the sympagic environment is therefore much increased. I would prefer to suspend any other judgment on the presented results until this issue is clarified.

Here follows a few specific comments linked to my argument expressed above.

L33-35     I would suggest the authors to further clarify this concept making clear that nutrient exchange is a combination of processes. One option would be to move the sentence that is now at lines 67- to here. The common interpretation of a process should come before the models and their parameterizations that approximate the real process to the best of their knowledge. The authors instead start by saying what models do, instead of saying what the nutrient exchange process at the sea ice-ocean interface entails.

L60-62     It is not clear whether $\Delta z$ is in the sea ice or in the ocean. I think this is the crucial point that I am addressing in the general comment.

L63     The expression "calculate tracer diffusion" is unclear. As suggested earlier, this should be one way of parameterizing the diffusion term in the overall mass-balance equation describing exchanges at the interface

L86-88     The parametrization proposed by Cota et al. (1987) is a formulation of Fick's law of diffusion. Nutrient exchanges are also due to diffusive processes. Cota's formulation is the boundary condition of any diffusive process modeled using a parabolic differential equation.

L71-74     It may be just a language issue, but this sentence seems to imply that nutrient availability in the sea-ice is mainly controlled by diffusive process. As recognized by the authors in their answer, this is just one of the components of nutrient exchange at the interface. Enhanced turbulent at the sea-ice bottom has the capacity to alleviate nutrient limitation in the absence of ice growth or melt.

L76-77     This is a rather bold statement. Is there any evidence that the relative change in the stock/rate associated to sea-ice primary production (that is only a fraction of the global ocean carbon flux) would lead to climatic feedback?

L79     I would suggest the authors to leave out the momentum flux, which is not parameterized the same way as heat and salinity (although based on the same arguments of Reynolds averaging).

L99     This parameterization is from McPhee et al (2008). The fact that it is implemented in CICE is secondary.

L120-121     I am not familiar with the Icepack notation, but I would suggest to use $z$ as the coordinate variable for the vertical rather than $x$.

---

## Author Response (AR2)

Dear Dr. Marcello Vichi,

Thank you very much for the time you took to review our manuscript and for your comments. In the following text we will address your comments and explain how they were incorporated into the revised manuscript or used to clarify the points you addressed.

We focus first on your general comment and then on your specific comments. Please note that when indicating line numbers where changes were implemented in the revised manuscript, we refer the version without tracked changes.

With my best regards,

Tromsø, 9 November 2021

(Pedro Manuel da Silva Duarte, on behalf of all co-authors)

**General comment**

In the first paragraph of your review, you write that many of your previous remarks were addressed in our previous review but that the work may be ameliorated, especially, in reducing the confronting tone used when referring to the previous literature. We did not mean to be "provocative" when addressing the literature but solely wish to emphasize why this study is important to conduct. To avoid a confrontational ton, we have removed Table 1, where several examples of models specifying the parameterizations used to compute diffusion of tracers were originally listed. Accordingly, the sentence in the Introduction where we referred Table 1 has become (lines 60-63):

"*The analysis of several models published over the last decades and their approaches to calculate tracer diffusion shows that some models do not consider this process or limit it to molecular diffusion. Other models consider turbulent exchanges parameterized as a function of the Rayleigh number, calculated from brine vertical density gradients. Only one of the sampled models (Mortenson et al., 2017) uses a parameterization based on friction velocity.*"

Regarding your main concerns about the paper that are expressed in the paragraphs to follow, we begin with one of your sentences in the 2$^{nd}$ paragraph:

"*My initial understanding was that the authors parameterized the turbulent flux at the interface with the seawater side.*"

Your initial understanding was, and it still is correct. We parameterized the turbulent flux solely at the interface between the bottom ice and the ocean.

In the 3$^{rd}$ paragraph you commented on the way we defined the terms on the left hand-side of equation 1. Please note that in the original we refer not to the term as a whole but to each of its components. For the sake of clarity, we have changed the text according to your suggestion.

You commented that extending the usage of equation 4 (3 in the revised manuscript) is not in accordance with McPhee but it is merely an extension of his concept. We have changed the text accordingly (line 93): *"This is an extension of the concept used for heat and salt by McPhee (2008)"*

In the same paragraph you noted the imprecision in using $N_i$, instead of $N_0$, which was also corrected in the revised manuscript (line 92).

In the next lines of the 3$^{rd}$ paragraph as well as in the 4$^{th}$ paragraph you develop further on the idea that we seem to have applied equation 4 (3 in the revised manuscript) within the brines. However, we did not do so, in line with our statement above confirming that the turbulent flux was parameterized solely at the interface between the bottom ice and the ocean. This misunderstanding was perhaps the result from the use of $h$ (the thickness of the biogeochemical grid in the CICE model) in equations 5, 6 and 7 (4, 5 and 6, respectively, in the revised manuscript). In the CICE model, the biogrid is vertically resolved in a number of layers. In our simulations we used 15 layers (please refer Table S3). Therefore, the existence of terms in the equations where the whole thickness of the biogrid ($h$) is used should not prevent the correct calculation of vertically resolved processes between the various layers, as well as between the bottom of the sea ice and the ocean. So, why then does this term show up? This happens because the biogrid is non-dimensional and the position of a point along the grid (described by x) is zero at the top of the biogrid and 1 at the ice-ocean interface. Therefore, in the CICE transport equation for the biogeochemical tracers the total thickness of the biogrid must always be multiplied by x or its differential to "convert" a relative distance to an "absolute" distance in meters. We added text explaining these details while citing the relevant references (please refer lines 96-116).

You commented that *"nowhere in the text it is stated that this is only applied at the boundary"*, referring to our turbulent diffusion parameterization. Please note that the following text to outline this approach is provided at the end of the Introduction (lines 69-71):

*"To test the above hypothesis, we use a 1D vertically resolved model and contrast results using the default diffusion parameterization and a "turbulent" parameterization analogous to that of momentum and heat transfer, at the interface between the ocean and the sea ice, based on McPhee (2008)."*

Please note also that before equation 10 in the previous version of the manuscript version, we wrote:

 *"We rewrite the last term of 8 **for the bottom ice layer** as:"*

In the revised manuscript we "reinforced" this statement in lines 120-123.

The 4$^{th}$ paragraph of your general comments begins with the following sentence:

*"What I find most confusing is the fact that eq (8) is valid within the brines, and the authors decided to change the diffusive term in a quite disputable way (that is, using a friction velocity that is only defined at the ocean water boundary)"*

Please note that equation 8 (7 in the revised manuscript) is valid within the brines until the sea ice bottom. However, this does not imply that it cannot include ice-ocean interface exchanges which in fact define the

bottom model boundary. Its last term includes molecular diffusion and mixed length diffusion which are calculated between each adjacent pair of layers and, in the last model layer, are calculated across the ocean-ice interface as well. We replaced the molecular diffusion component with turbulent diffusion only across the ice-ocean interface. We have removed equation 10 as it is no longer necessary after the following sentence we added in its place (lines 120-123):

*"The last term of equation 7 includes the contribution of molecular diffusion that is calculated at the interface of all layers of the biogrid and at the interface of the last layer and the ocean. In the simulations using turbulent diffusion, we perform the same calculations, except that the molecular diffusion term $\frac{\varphi D_m}{h^2}$ is replaced with a turbulent diffusion term $\frac{\alpha_s u^*}{h}$ at the interface between the last model layer and the ocean."*

In summary, we hope that our changes to the manuscript and responses have clarified that: (i) we did not assume turbulent diffusion within the brines was identical to the one in the ocean boundary layer; (ii) that we did not inappropriately handle the equations.

We removed the example of turbulent diffusion based on Cota et al. (1987) because, after reading more carefully the cited paper, we realized that turbulent diffusion was used in that study to calculate nutrient replenishment of the mixed layer and not exchanges across the ocean-ice interface. This has no influence on results or discussion presented.

We also removed most of the references to momentum exchanges following your suggestion.

**Specific comments**

L33-35 I would suggest the authors to further clarify this concept making clear that nutrient exchange is a combination of processes. One option would be to move the sentence that is now at lines 67- to here. The common interpretation of a process should come before the models and their parameterizations that approximate the real process to the best of their knowledge. The authors instead start by saying what models do, instead of saying what the nutrient exchange process at the sea ice-ocean interface entails.

**Answer**: Done as suggested. Please refer lines 25-28.

L60-62 It is not clear whether $\Delta z$ is in the sea ice or in the ocean. I think this is the crucial point that I am addressing in the general comment.

**Answer**: As outlined above, we removed this equation in the revised manuscript and hopefully with the changes described before this misunderstanding is resolved.

L63 The expression "calculate tracer diffusion" is unclear. As suggested earlier, this should be one way of parameterizing the diffusion term in the overall mass-balance equation describing exchanges at the interface.

**Answer**: We rephrased this to "calculate tracer diffusion across the ice-ocean interface" (line 60).

L86-88 The parametrization proposed by Cota et al. (1987) is a formulation of Fick's law of diffusion. Nutrient exchanges are also due to diffusive processes. Cota's formulation is the boundary condition of any diffusive process modeled using a parabolic differential equation.

**Answer**: This example was removed from the paper for reasons explained above.

L71-74 It may be just a language issue, but this sentence seems to imply that nutrient availability in the sea-ice is mainly controlled by diffusive process. As recognized by the authors in their answer, this is just one of the components of nutrient exchange at the interface. Enhanced turbulent at the sea-ice bottom has the capacity to alleviate nutrient limitation in the absence of ice growth or melt.

**Answer:** We write *"when brine gravity drainage is limited"* which implies we acknowledge other mechanisms stressed in the first paragraphs of the Introduction (line 64).

L76-77 This is a rather bold statement. Is there any evidence that the relative change in the stock/rate associated to sea-ice primary production (that is only a fraction of the global ocean carbon flux) would lead to climatic feedback?

**Answer**: The sentence was removed.

L79 I would suggest the authors to leave out the momentum flux, which is not parameterized the same way as heat and salinity (although based on the same arguments of Reynolds averaging).

**Answer:** Done as suggested.

L99 This parameterization is from McPhee et al (2008). The fact that it is implemented in CICE is secondary.

**Answer**: The authorship of McPhee is acknowledged in the sentence. The reason we include the equation is to emphasize that the approach we adapted for nutrients is already implemented for heat in the CICE model. This is to reinforce our arguments about using consistent approaches to heat and dissolved tracers.

L120-121 I am not familiar with the Icepack notation, but I would suggest to use z as the coordinate variable for the vertical rather than x.

**Answer**: z is the vertical distance; x is the relative distance. So, we think z is used as you suggest. Here we followed exactly the notation used in the cited references for the sake of clarity.

Dear Dr. Martin Vancoppenolle,

Thank you very much for the time you took to review our manuscript and for your encouraging comments.

Please note that in the revised version we removed the previous Table 1, following your suggestion. We made some changes in the manuscript following the comments from Dr. Marcello Vichi and tried to clarify better a few issues. These changes have no implications in the paper concepts, results or conclusions.

With my best regards,

Tromsø, 9 November 2021

(Pedro Manuel da Silva Duarte, on behalf of all co-authors)

---

## Author Response (AR3)

Dear Dr. Marcello Vichi,

Thank you very much for the time you took to review our manuscript and for your comments. In the following text we will address your comments and explain how they were incorporated into the revised manuscript or used to clarify the points you addressed. When referring to line numbers we consider the reviewed manuscript without tracked changes.

With my best regards,

21 December 2021

(Pedro Manuel da Silva Duarte, on behalf of all co-authors)

**Comments and answers**

**Comment**

"At line 87-89 the authors should make explicit that this testing is specifically applied to the CICE model and its technical features. The implementation may be different in another model, and not even be appropriate in certain cases. This is necessary because the theory is presented as grounded on the specific configuration and parameters chosen by CICE and ICEPACK (e.g. line 109)."

**Answer**

Done as suggested. Please refer lines 76-77 of the revised manuscript.

**Comment**

"1. I think the comparison with and reference to the heat flux is misleading and should be removed. It can be done in the introduction and brought back in the discussion, but in the conceptualization phase is confusing and not entirely justified. Starting from the case of salinity is more adequate since nutrients are solutes just like salt. Like at line 99, the authors could say: "When this quantity is salt in seawater, the formulation is:" Heat is also mentioned at lines 105-106 while only salinity has been presented in the previous sentences. This also also applies to the turbulent exchange coefficients alpha. I would suggest the authors to just refer to \alpha_s and its range of variability, and not to \alpha_h and how it relates to salt exchange.

In particular, I would kindly request the author to revise their statement that the material exchange fluxes should be made equivalent to the treatment of the heat fluxes, because they are actually resolved in different ways in the CICE implementation, and moreover, such a statement is not of general applicability to all models."

**Answer**

Done as suggested. We also removed the equation for the heat flux and the text about the relationship between the heat and the salt transfer coefficients, from section 2.1.

**Comment**

"2. The most important change is to add the explicit parameterization of the bottom boundary condition before explaining the time scales (L116-L137). I would recommend the authors to first present eq. 7 and then the boundary condition that is resolved by CICE (eq. 40 in Jeffery et al. 2016). At this point, they should clarify their proposed parameterization of the boundary flux and how they modified the implementation of the diffusivity at this boundary grid point. Please make sure to indicate that this point is an interface grid point representing ocean conditions, with porosity equal to 1.

Finally, I would kindly request the authors in their discussion to clarify that this proposed parameterization is based on an approximation of the molecular sublayer and the related physical processes. That specific measurements are required to further improve its description and generalization for applications to other model formulations."

**Answer**

Section 2.1 was reformulated. We reordered the equations, following the recommendations from the referee. Please refer lines 92-117. Moreover, we added a new paragraph to the end of the Discussion section (lines 442-459) where we addressed some of the concerns expressed by the referee.

We also added a few sentences to the Introduction (lines 63-67) to better emphasize comparable approaches carried out in previous studies.

---

## Author Response (AR4)

Dear Dr. Guy Munhoven,

Thank you very much for the time you took to review our manuscript. We corrected the text following all technical corrections pointed out in your revision.

With my best regards,

3 January 2022

(Pedro Manuel da Silva Duarte, on behalf of all co-authors)